



# Evaluation of regional isoprene emission factors and modeled fluxes in California

Pawel K. Misztal[1,2], Jeremy C. Avise[3,4], Thomas Karl[5], Klaus Scott[3], Haflidi H. Jonsson[6], Alex B. Guenther[2,4,7], and Allen H. Goldstein[1]

[1]University of California at Berkeley, Berkeley, California, USA
[2]National Center for Atmospheric Research, Boulder, Colorado, USA
[3]California Air Resources Board, Sacramento, California, USA
[4]Washington State University, Department of Civil and Environmental Engineering, Pullman, Washington, USA
[5]Institute for Meteorology and Geophysics, University of Innsbruck, Innsbruck, Austria
[6]Center for Interdisciplinary Remotely-Piloted Aircraft Studies, Monterey, California, USA
[7]Department of Earth System Science, University of California, Irvine, California, USA

*Correspondence to:* P. K. Misztal (pkm@berkeley.edu)

**Abstract.** Accurately modeled Biogenic Volatile Organic Compound (BVOC) emissions are an essential input to atmospheric

chemistry simulations of ozone and particle formation. BVOC emission models rely on Basal Emission Factors (BEFs) distribution maps based on emission measurements and vegetation landcover data but these critical input components of the models as well as model simulations lack validation by regional scale measurements.

We directly assess isoprene emission-factor distribution databases for BVOC emission models by deriving BEFs from direct airborne eddy covariance (AEC) fluxes (Misztal et al., 2014) scaled to the surface and normalized by the activity factor of

10 the Guenther et al. (2006) algorithm. The available airborne BEF data from approx. 10,000 km of flight tracks over California were averaged spatially over 48 defined ecological zones called ecoregions. Consistently, BEFs used by three different emission models were averaged over the same ecoregions for quantitative evaluation. Ecoregion-averaged BEFs from the most current landcover used by the Model of Emissions of Gases and Aerosols from Nature (MEGAN) v.2.1 resulted in the best agreement among the tested landcovers and agreed within 10% with BEFs inferred from measurement. However, the correlation was

15 sensitive to a few discrepancies (either overestimation or underestimation) in those ecoregions where landcover BEFs are less accurate or less representative for the flight track. The two other landcovers demonstrated similar agreement (within 30% of measurements) for total average BEF across all tested ecoregions but there were a larger number of specific ecoregions that had poor agreement with the observations.

Independently, we performed evaluation of the new California Air Resources Board (CARB) hybrid model by directly

comparing its simulated isoprene area emissions averaged for the same flight times and flux footprints as actual measured area emissions. The model simulation and the observed surface area emissions agreed on average within 20%.

We show that the choice of model landcover input data has the most critical influence on model-measurement agreement and the uncertainty in meteorology inputs has a lesser impact at scales relevant to regional air quality modeling.





## 1   Introduction

Vegetation in California emits isoprene, terpenes, and oxygenated biogenic volatile organic compounds (BVOC) which react with anthropogenic pollutants to form ozone and particulate matter. Isoprene (2-methyl-1,3-butadiene) is the dominantly emitted BVOC globally (Guenther et al., 2012; Sindelarova et al., 2014) and the single most important species affecting regional air quality in most regions (Unger et al., 2013; Müller et al., 2008; Henze and Seinfeld, 2006; Rosenstiel et al., 2003) including California.

Based on previous BVOC emission measurements from Californian oak woodlands, which were made exclusively at branch and leaf levels (e.g. Winer et al., 1992), the vast majority of California's isoprene emissions are expected to occur from oak trees and to some extent from Eucalyptus trees. The dominant oak environments in California are located in the foothills encompassing the Central Valley and along the Pacific Coast Ranges. Previous studies have shown that estimation of biogenic emissions is uncertain because of the lack of regional-scale measurements and differences in driving input variables as well as the way the model components are calculated. Guenther et al. (2006) and Arneth et al. (2011) presented the sensitivity of BVOC emission estimates to landcover and weather/climate variables. Other parameters related to the driving inputs such as spatial (Pugh et al., 2013) or temporal (Ashworth et al., 2010) resolutions have also been shown to impact MEGAN model performance. Situ et al. (2014) performed a detailed study of the importance of input variables and parameters on emissions simulated by the MEGAN model using a Monte Carlo approach and suggested that large uncertainties of emission estimates can be reduced if emission factor, photosynthetically active radiation (PAR) and temperature input accuracies are improved. There are currently no algorithms for modeling accurately the emission response to stresses (e.g. water stress) which requires further mechanistic understanding of biogenic emissions and more ecosystem-scale measurements (Potosnak et al., 2014).

Despite the knowledge of complexities behind accurate modeling, without regional measurements there is no reliable means of verifying whether modeling simulations of biogenic emissions and air quality work well across the specific regions. Recently, direct airborne eddy covariance (AEC) measurements based on continuous wavelet transformation have become a valuable tool for quantifying emission sources and sinks of atmospheric reactive gases (Misztal et al., 2014; Yuan et al., 2015; Wolfe et al., 2015) and these types of measurements are uniquely valuable for validation of the regional biogenic emission models and landcover emission factor driving inputs.

The California Airborne BVOC Emission Research in Natural Ecosystems Transects (CABERNET) study was conducted in early summer 2011 to directly measure for the first time the regional scale BVOC emissions using an aircraft with one of the goals being evaluation of the performance of the emission models used by California Air Resources Board (CARB) in simulating state-wide air quality. Eight research flights were conducted including mostly horizontal transects (Misztal et al., 2014) to measure the regional emissions over the majority of oak woodland regions in California at a 2-km spatial resolution. In addition, stacked gradient profiles were flown at multiple altitudes to measure vertical flux divergence (Karl et al., 2013) allowing scaling of aircraft-level flux measurements to ground-level emissions (surface emissions). We flew most extensively over areas identified as code 6 (Central California Foothills and Coastal Mountains) in the level III United States Environmental Protection Agency (USEPA) ecoregion classification(USEPA, 2014) (see USEPA ecoregion map in Figure 1). The 29 sub-





ecoregions (level IV) of the level III ecoregion 6 comprise oak woodlands which were confirmed to be dominant isoprene emission sources with effective measured basal emission factors (BEFs) of more than 4 mg m$^{-2}$ h$^{-1}$ and occasionally up to around 10 mg m$^{-2}$ h$^{-1}$ (Misztal et al., 2014). Large changes in temperatures (and radiation) during the field campaign as well as the preceding meteorological history were responsible for a broad range of observed emissions from less than 1 mg m$^{-2}$

h$^{-1}$ on a cool day to about 15 mg m$^{-2}$ h$^{-1}$ (or more) on a hot day over a densely populated oak area (Misztal et al., 2014).

Here we use our previously published direct airborne flux measurements to infer isoprene BEFs (referred to as measured BEFs) to evaluate emission factors based on landcovers (referred to as landcover BEFs) used by the three models typically applied in California: 1) Biogenic Emission Inventory processing model (BEIGIS) (Scott and Benjamin, 2003), 2) Model of Emissions of Gases and Aerosols from Nature (MEGAN) v.2.04 (Guenther et al., 2006), and 3) MEGAN v.2.1 (Guenther

et al., 2012). Independently, we evaluate performance of the new California Air Resources Board (CARB) model (MEGAN v.2.04 and BEIGIS hybrid which included enhancements from MEGAN v.2.1) by directly comparing simulated isoprene area emissions averaged for the same flight times and flux footprints as actual measured area emissions.

## 2 Methods

### 2.1 Modeling approaches

Modeling of BVOC emissions involves a framework including emission factors, short-term and long-term emission algorithms and a canopy environment model (a model to relate above canopy environment to leaf level conditions), along with data bases of landcover and meteorological driving variables.

Different models use often different inputs to simulate isoprene emissions and each model is characterized by its specific architecture. The following models are commonly used for simulating biogenic emissions in California: 1) BEIGIS (CARB's

original biogenics model) using the US Geological Survey's Gap Analysis Project (GAP) landcover database to quantify coverage of oaks and other species composition (Scott and Benjamin, 2003; Davis et al., 1998), 2) MEGAN v.2.04, landcover v.2.1 (Guenther et al., 2006) based on WestGAP landcover database and Forest Inventory and Analysis (FIA) National Program, and 3) MEGAN v.2.1, landcover v.2.2 (Guenther et al., 2012) – based on the National Landcover Dataset (NLCD, Homer et al.(2004)), FIA, and plant functional type (PFT) datasets.

MEGAN v.2.1 model provides the most current and accurate landcover, but the model architecture is not significantly different from MEGAN v.2.04 for isoprene. BEIGIS model shares MEGAN v.2.04 architecture but uses different landcover and vegetation specific emission factors. Following the CABERNET measurements, further enhancements from MEGAN v.2.1. were adopted by CARB resulting in a development of a hybrid BEIGIS/MEGANv.2.04/v.2.1 model designed for regional simulations, and its statewide emission estimates of isoprene are evaluated here with CABERNET measured AEC fluxes.

The three model architectures are extremely similar because they evolved from the same roots. Differences between the model outputs occur mainly due to differences in the landcover driving variables (plant species composition, leaf area index (LAI)) and meteorological driving variables (light, temperature). When comparing different models with observations, it is important to first determine the effects of different input variables that are used and perform extensive sensitivity studies. The



resolution and evaluation of these driving variable databases is particularly critical in the areas close to the mountains that typically have high gradients of temperature and vegetation and where meteorological stations may not be as densely spaced compared to near the urban areas or where gradients in temperature are smaller. Since the models predict that the major isoprene source regions in California are predominantly oak savannas in the foothills where temperature estimates are uncertain, this
can contribute uncertainties in isoprene emission estimates.

To evaluate the accuracy of the landcover used as the basis for the models' emission factor distributions, we used the 2-km resolution measured flux data normalized for temperature and PAR according to the Guenther et al. (2006) algorithm to derive airborne BEFs. The inverse emission algorithm approach has been used earlier at a canopy scale (Misztal et al., 2011) and recently to derive BEFs from satellite measurements of formaldehyde (Marais et al., 2014). To evaluate the meteorological
driving variables, we compared hourly temperature data simulated by the Weather Research and Forecasting (WRF) model (Skamarock et al., 2005) at a 4x4 km resolution with available weather station data along some of the CABERNET flight tracks.

### 2.1.1 BEIGIS

The Biogenic Emission Inventory processing model (BEIGIS) (Scott and Benjamin, 2003) was developed by CARB. BEIGIS
uses California landcover, leaf mass, and emission rate databases with a geographic information system (GIS), is a regional model specific to California, and is spatially resolved at 1 $km^2$ and temporally at 1 hour. The initial set of BEIGIS inputs includes GIS-based maps of landcover types. They are based on a USGS Gap Analysis Project (GAP) biodiversity database which covers natural areas of California (Scott et al., 1993; Davis, 1994; Karlik et al., 2003). The database was generated from summer 1990 Landsat Thematic Mapper satellite images, 1990 high altitude color infrared imagery, vegetation maps
based on historical field surveys, and other miscellaneous vegetation maps and ground surveys. The urban and crop areas are not represented by the GAP database and use independent maps. These maps are subsequently used to assign mostly branch-scale emission factors which in the case of GAP covered areas come from a compilation by Benjamin et al. (1996) and a specific leaf weight (to convert LAI to biomass density) database (Nowak et al., 2000). The landscape emission factor layers are subsequently formed and are used with environmental correction algorithms Guenther et al. (1993); Harley et al.
(1998) using hourly temperature and solar radiation datasets gridded at 4 $km^2$. A canopy environment model is not used in BEIGIS, and it is assumed that the branch-scale emission factors account for shading and canopy environment effects. The model has many similarities to the predecessor of the MEGAN model (Guenther et al., 1993, 1995) since it is using similarly derived emission factor maps (GAP/FIA, branch-scale emission factors) and a similar framework for application of light and temperature algorithms, except that the BEIGIS model was specifically optimized for California. This includes using an 8-
30   day LAI and phenology database, where specific phenology masks are applied to deciduous trees and shrubs, grasses and herbaceous plants to turn on and off their emissions at different times of year, while evergreen trees and some shrubs are assumed to have emissions all year.





### 2.1.2  MEGAN v.2.04

The Model of Emissions of Gases and Aerosols from Nature (MEGAN) v.2.04 (Guenther et al., 2006) was used in the initial stages of our study to plan CABERNET flight tracks and was also tested in the early stages of measurement model comparisons using the observed airborne BEFs. MEGAN is designed for both global and regional emission modeling with 1 km2 spatial

resolution. This version of MEGAN defined emission factors as the net flux of a compound into the atmosphere which was intended to account for losses of primary emissions on their way into the above canopy atmosphere. The model uses an approach that divides the surface of each grid cell into different Plant Functional Types (PFTs) and non-vegetated surface. The PFT approach enables the MEGAN canopy environment model to simulate different light and temperature distributions for different canopy types (e.g., broadleaf trees and needle trees). In addition, PFTs can have different LAI and leaf age

seasonal patterns (e.g., evergreen and deciduous). MEGAN v.2.04 accounts for regional variations using geographically gridded databases of emission factors for each PFT. The standard MEGAN global classification included 7 PFTs, but for regional modeling a classification scheme can have any number of PFTs.

### 2.1.3  MEGAN v.2.1

The MEGAN v.2.1 model (Guenther et al., 2012) includes enhancements to MEGAN v.2.04. The main architecture of the

model is very similar (see the Supplement Fig. S1) but there are several significant differences in how emission factors are represented, deposition to the leaf surface accounted for (relevant for species such as methanol but not isoprene), more generic PFTs are used for global modeling, and most importantly a new landcover database (v.2.2) is included that was derived by combining high resolution imagery (60 m, and 30 m) with species composition data. The base MEGAN v.2.1 landcover v.2.2 includes more than 2000 ecoregions which allows for the emission factor for a given PFT (e.g. temperate needleleaf trees)

to change as a function of ecoregion. The MEGAN landcover product is further described in "Landcovers" section below. While the previous version of MEGAN (v.2.0) defined emission factors as the net flux of a compound into the atmosphere, the MEGAN (v.2.1) emission factor represents the net primary emission that escapes into the atmosphere but is not the net flux because it does not include the flux of chemicals from the above canopy atmosphere down into the canopy. Emission factors based on scaled up leaf level emissions inherently exclude the deposition component. In order to use above canopy flux

measurements to establish emission factors, an estimate of the deposition flux is added to the above canopy flux measurements to determine the MEGAN v.2.1 emission factors. For isoprene this deposition flux estimate is equal to zero.

### 2.1.4  CARB's hybrid model

The MEGAN v.2.04 model framework was adapted at CARB to include MEGANv.2.1 enhancements such as 8-day LAI (as opposed to monthly average LAI), longer-term (10-day) temperature and PAR impacts on the emission (consistent with Guen-

ther et al. (2006) algorithm), and many of the California specific datasets developed in conjunction with the BEIGIS model. For this study, the model was run at 2 km x 2 km resolution and driven by meteorology at 4 km x 4 km. The LAI data used was the 8-day MODIS LAI for 2011. This regional model most closely agreed with the measured fluxes and is also currently used





by CARB to estimate the BVOC emissions inventory for California. While we show BEF comparison for all three model's landcovers, we narrow our model comparison to the CARB's hybrid model. In this application of MEGAN (v.2.04), the model produced hourly emissions estimates at a 2 km x 2 km resolution. To facilitate the model – measurement comparison, the hourly emission estimates were interpolated to the measurement time stamps and the modeled flux was calculated in a GIS

environment as follows: 1) convert the grid cell emission rates to areal fluxes; 2) calculate the area weighted average flux (based on intersecting the grid with the flux footprint); and 3) convert the area weighted flux to an emission rate by multiplying by the calculated footprint area.

The flux footprint corresponding to each aircraft measurement is calculated as the half-width of the Gaussian distribution, which accounts for 90% of the total flux. In order to account for the remaining 10% of the flux, an additional 10% is added to

the simulated area weighted emissions.

## 2.2   Model domain and ecoregions

The CABERNET flights covered a large portion of California including representative areas with high densities of oak trees which are expected to dominate the statewide isoprene emissions. Ecoregions denote areas of general similarity in ecosystems and in the type, quality, and quantity of environmental resources (Griffith et al., 2008).

A map of California ecoregions overlaid with the CABERNET flight tracks (shown earlier in Figure 1) provides information on the extent of their spatial coverage with respect to airborne measurements. Most of the subecoregions (level IV) belonging to the ecoregion 6 (level III:Central California Foothills and Coastal Mountains) denoted in yellow were covered, as well as some subecoregions of the ecoregion 7 (Central California Valley) in brown, ecoregion 5 (Sierra Nevada) in green, and ecoregion 14 (Mojave Basin and Range) in pink. Of the 48 subecoregions flown over during the CABERNET campaign, 29 subecoregions

were within ecoregion 6 which comprises most of the oak woodlands in California.

The primary distinguishing characteristic of ecoregion 6 is its Mediterranean climate of hot dry summers and cool moist winters, and associated vegetative cover comprised mainly of isoprene emitting oak woodlands. Ecoregion 6 also includes non/low- isoprene emitting chaparral and grasslands which occur in some lower elevations and patches of pine are found at the higher elevations. Surrounding the lower and flatter Central California Valley (ecoregion 7), most of the region consists of

open low mountains or foothills, but there are some areas of irregular plains and some narrow valleys. Large areas in ecoregion 7 are used as ranch lands and grazed by domestic livestock. Relatively little land in this ecoregion has been cultivated, although some valleys are major agricultural centers such as the Salinas area or the wine vineyard centers of Napa and Sonoma. Natural vegetation includes coast live oak woodlands, Coulter pine, unique native stands of Monterey pine in the west, and blue oak, black oak, and grey pine woodlands to the east (USEPA, 2014).





### 2.3   Driving inputs

#### 2.3.1   Landcovers

The landcover used to drive the model has a critical influence on model performance because it defines the type of vegetation or plant function type (PFT), land fraction, and finally determines the emission factor. Up-to-date landcover products should give

more accurate results because the landcover can change due to growing and senescing vegetation, fires, and land-use change or plant species composition change. The airborne flux measurement-model comparison provides an opportunity to identify any inaccuracies in landcover databases which can then be used to improve them. Landcovers used by the models in this study are presented in Figure 2.

The Gap Analysis Program (GAP) database can be used to construct the spatial distribution of oak woodland areas (Figure

2a). This distribution is extremely similar to the BEIGIS emission factors (Figure 2b) which were based on the GAP data. While the global MEGAN v.2.04 landcover v.2.1 (Figure 2c) was also based on FIA and WestGAP datasets and interestingly showed almost identical BEF means for isoprene compared to BEIGIS isoprene BEFs, the standard deviations of spatial variability were much different with BEF distribution that were more smoothed out across many areas of California. The latest MEGAN v.2.1 landcover v.2.2 (Figure 2d) is a state-of-the-art product which showed the most accurate match with airborne fluxes.

This landcover is based on a high resolution (60 m) PFT database using the Community Land Model 4 (CLM4) PFT scheme generated for the US for the year 2008 and is available with the MEGAN v.2.1 input data (http://bai.acd.ucar.edu/MEGAN/) (Guenther et al., 2012). The database was created by combining the National Landcover Dataset (NLCD, Homer et al. (2004)) and the Cropland Data Layer (see http://nassgeodata.gmu.edu/CropScape/), which are based on 30-m LANDSAT-TM satellite data, with vegetation species composition data from the Forest Inventory and Analysis (www.fia.fs.fed.us ) and the soil database

of the Natural Resources Conservation Services (http://sdmdataaccess.nrcs.usda.gov/). The processing included adjusting the NLCD tree cover estimates in urban areas to account for the substantial underestimation of the LANDSAT-TM data (Duhl et al., 2011). The California Information Node (CAIN) database from the UC Davis repository (http://ice.ucdavis.edu/project/cain) contains exactly the same habitats as the GAP database but was independently derived. The CAIN database augmented several datasets linked to the National Biological Information Infrastructure (NBII) which was linked to the California Department of

Forestry and Fire Protection (CalFire) Fire and Resource and Assessment Program (FRAP). This database was also based on the FIA, and complements the GAP database, in particular in southern CA. The northwest region of CA is more extensively represented by GAP. Combination of the GAP and CAIN dataset therefore is useful in the context of BVOC emission modeling in California.

#### 2.3.2   Temperature and radiation

Hourly temperature data were simulated by WRF at 4 km x 4 km resolution. Based on comparison with weather station close to gradient stacked profile in RF6 and RF7, we found that WRF spatial resolutions lower than 8 km x 8 km can lead to temperature inaccuracies of more than 3 °C during peak periods (Figure 3). Similar conclusions were made by Yver et al. (2013). For validation of WRF temperature data a diagnostic meteorological model (CALMET) was used by CARB. Despite mostly good





agreement, areas were identified with large discrepancies. Since CALMET interpolates in 2D the temperature surface from the available met stations, inaccuracies may be expected in areas were stations are not densely represented. The optimal approaches for California were found to be the 4 x 4 km WRF model nudged by CALMET or CALMET directly. The dynamics of the temperature changes close to the foothills during a day can be seen on the animation (http://tinyurl.com/wrftempcabernet) where gradients are very high.

Photosynthetically Active Radiation satellite datasets were recently validated by Wang et al. (2011) and Guenther et al. (2012). The CARB's model (adapted MEGAN application) used the WRF insolation directly.

### 2.3.3 LAI

The LAI dataset used was the current LAI from MODIS for the flight days and CARB's LAI data was the Terra/Aqua combined 8-day product.

### 2.4 CABERNET direct flux dataset

Detailed description of the campaign's 8 research flights (RFs) can be found in Karl et al. (2013) and Misztal et al. (2014). The airborne fluxes which were reported in Misztal et al. (2014) were subsequently processed using the inverse of the Guenther et al. (2006) algorithm (Eq. 1) into: 1) airborne Basal Emission Factors (BEFs) and 2) spatially averaged gridded emissions using the flux footprints. More methodological details are provided in the Supplement.

### 2.4.1 Application of inverse G06 algorithm to the airborne fluxes

Comparison of the measured fluxes to the model emission potentials was done after calculating BEFs from the measurements. The raw data undergoes the following workflow to obtain airborne BEFs from the airborne fluxes: 1) Application of wind corrections from "Lenschow maneuvers"; 2) Derivation of airborne concentrations from daily calibrations; 3) Wavelet and FFT flux derivation at aircraft altitude; 4) Interpolation of fluxes at aircraft altitude to the surface fluxes using coefficients from racetracks, and the ratio of the altitude above the ground (z) to planetary boundary layer depth (zi) (i.e. accounting for flux divergence); 5) Spatial averaging of surface fluxes to 2 km resolution; and 6) Derivation of BERs by normalization of the surface fluxes using surface temperature and PAR according to MEGAN algorithm which accounts for previous temperature and PAR history (equation from Misztal et al. (2011)):

$$\mathrm{BEF_{AEC}} = \frac{F_{\mathrm{AEC}}}{\gamma_{T,\mathrm{PAR}}}, \tag{1}$$

where BEF is airborne basal emission factor, and $\gamma_{T,\mathrm{PAR}}$ is the Guenther et al. (2006) algorithm's activity factor which accounted for temperature ($T$) and PAR of the current hour, as well as the $T$ and PAR averaged over the previous 24 and 240 hours. Further details on the application of the inverse algorithm can be found in the Supplement.





### 2.4.2 Flux footprint application

The footprint for each flux point was derived using the Weil and Horst (1992) approach and depends on the wind speed, relative altitude to the PBL height, and the convective velocity scale. Here we use scaling developed for the mixed layer according to:

$$dx_{0.5} = 0.9 \cdot \frac{u \cdot z_{\mathrm{m}}^{2/3} \cdot h^{1/3}}{w^*},\tag{2}$$

where $dx_{0.5}$ is the half width of the horizontal footprint, $u$ the horizontal windspeed, $z_{\mathrm{m}}$ the height above ground, $h$ the PBL height and $w^*$ the convective velocity scale which is derived from the wavelet heat flux in each transect.

The source contribution area can be approximated by projecting an upwind-pointed half dome with the dx0.5 parameter representing a radius of that half dome. As an example this leads to a footprint of 3.1 km for h=2000 m, zm=1500 m, u = 3.5 m/s and w* = 1.7 m/s encountered during RF6. The upwind fetch was on the order of 12 km for RF6 and RF7. The footprint

is represented by the half-widths which can be regarded as a distance between the points of the Gaussian curve where the flux falls to the half of its maximum. Therefore, the flux contribution is not the same within the halfwidth. The area of such a footprint is approximately 90% of the flux contribution relative to the entire footprint (the full Gaussian). This approximation assumes a symmetrical footprint, but in reality the footprint area is larger along the direction that the wind is blowing.

## 3   Results and discussion

### 3.1   Landcover - a critical driving variable

The driving variables used in the models are much more important for prediction accuracy than the different model architectures. This observation is consistent with reports comparing different process-based models which differ in the modeling framework but give similar estimates when exactly the same input variables are used (Arneth et al., 2011). For example, Ashworth et al. (2010) used MEGAN to evaluate how sensitive isoprene emissions are to different time resolutions of the input

data and showed that even a 70% underestimation can result from using overly coarse data. Detailed descriptions for each of the input variables tested are shown in the Supplement. We draw particular attention to landcover emission factors used by the MEGAN v.2.04, MEGAN v.2.1 and BEIGIS models, because they showed significant regional discrepancies despite having similar state-wide averages. To demonstrate where exactly these quantitative differences exist, the emission factors from landcovers used by BEIGIS and MEGAN v.2.04 were subtracted from the most current landcover used by MEGAN v.2.1

which served as a reference (Figure 4). The green areas in Figure 4 denote those areas where absolute agreement between the landcover BEFs was within $\pm 0.5$ mg m$^{-2}$ h$^{-1}$. These areas occupy more than half of California, but they are mostly where absolute isoprene emission strengths are low (Central Valley, Mojave Desert, etc.). The largest negative differences for both MEGAN v.2.04 and BEIGIS landcovers are observed in the oak woodland areas surrounding the Central Valley of California. The BEIGIS landcover highest emission factors are correctly concentrated over the oak bands but their absolute magnitude was

higher than in MEGAN v.2.1 landcover with differences sometimes exceeding 10 mg m$^{-2}$ h$^{-1}$. In contrast, the MEGAN v.2.04 landcover had positive differences in the Sierra Mountains and close to the coast. The distribution of maximal emission factors



is often offset in the models as in the MEGAN v2.04 landcover where BEFs are more smoothly dispersed and extend over part of the Central Valley as well as in the coniferous areas on the mountains where isoprene should be low. This is again in contrast to BEIGIS landcover where the BEFs change more sharply from very low to very high and vice versa. These landcovers are later quantitatively compared with airborne BEFs.

## 3.2 Comparison of MEGAN v.2.1 landcover v.2.2 BEFs to airborne BEFs

### 3.2.1 2-km BEFs

Isoprene emission model estimates were based on landcover basal emission factors, landcover distributions, and the changes in emission associated with the environmental parameters temperature and PAR. Measured AEC fluxes scaled to the surface and normalized for temperature and radiation using the Guenther et al. (2006) activity factor to derive airborne BEFs were directly compared to emission factors used by the three different models. A spatial map of measured BEFs at 2 km was overlaid over BEFs from the latest MEGAN v.2.1 landcover v.2.2 (Figure 5).

This comparison approach has some uncertainty due to the temperature and PAR datasets and the algorithm used for calculating the activity coefficient, which are much larger than the uncertainty of the measured surface fluxes because of high sensitivity to errors in temperature and PAR. However, this approach is useful because we can compare the measured BEF (essentially the measured emission potential for that ecosystem) to the BEF used to drive the model for that ecosystem. The spatial comparison clearly shows a remarkable correspondence between airborne BEFs derived at 2 km spatial resolution with landcover BEFs at a similar resolution. The transition from the low emitting environment in the Central Valley to highly emitting areas occupied by oak woodlands is clear. The most accurate matches can be seen, for example, in the central part of the Sierra foothills and on the southern Coastal Range, to the south east of Monterey Bay and in the oak savannas near San Francisco Bay (East Bay hills, and Diablo Valley). The BEFs decline to zero over water bodies (e.g. San Francisco Bay, or lakes in the central-northern Sierras). There are some areas which do not agree well, for example, in the north-east over the Sierras which is dominated by conifers where airborne BEFs were somewhat lower than predicted. On the other hand, there are areas where the aircraft observed higher BEFs (e.g. beginning of the Central Coastal Range track south of the Monterey Bay in the 6ag ecoregion) that are most likely related to inaccuracies in the oak landcover database and to a lesser degree could come from potential PAR/temperature bias.

### 3.2.2 Eco-region specific evaluation of BEFs

California landscapes differ sustantially in plant species composition, plant functional types, and fractional coverage of vegetation. It therefore makes sense to look at model-observation comparisons separately for distinct ecological zones. We flew over 48 distinct subecoregions (level IV) which constitute more than a quarter of California ecoregions covering 120,000 km$^2$ which is 29% of the area of California. These subecoregions are nested within 4 broader ecoregions (level III). Ecoregion 6 comprises most of the oak woodlands in the Central California Foothills and Coastal Mountains, and we flew over 29 of its 44 subecoregions (6a-6ar). Ecoregion 7 is characterized by very low isoprene emission potential and includes most of the





Central California Valley, and we flew over 14 of its subecoregions. We also transected 2 subecoregions of the Sierra Nevada (ecoregion 5) and 3 of the Mojave Basin and Range (ecoregion 14).

The measured isoprene BEFs were much higher over ecoregions 5 and 6 than over ecoregions 7 and 14. Within ecoregion 6's subecoregions there was significant variability of BEFs ranging from near zero to above 10 mg m$^{-2}$ h$^{-1}$. The BEFs from the MEGAN v.2.1 landcover v.2.2 in most cases fell in the same range as measured BEFs, but in some cases they were higher. The landcover BEF means are the averages of the entire area of each ecoregion while measured BEFs represent only the part of those areas where CABERNET flights were done. This could be particularly important for the Sierra foothills where the footprint was often overlapping with the less dense portions of the oaks in the lower part of the foothills, and therefore may not be representative of the subecoregion average. Comparison of the measured versus modeled emissions integrated over the same flux footprint areas are shown later. Nevertheless, this BEF comparison is independent of the footprint calculation and is indicative of the relatively good agreement we observed between measured and modeled isoprene emissions for most ecoregions.

Using a scatter plot of average modeled versus measured BEFs (Figure 6), it is possible to assess if the model's landcover input does a reasonable job over each of these different ecoregions. MEGAN v.2.1 Landover v.2.2 resulted in the smallest number of outlying ecoregions and overall showed the best fit.

Statistics needs to include the outliers but it is interesting also to evaluate the influence of outliers on the fits of the measured BEF with each model. Inaccuracies in the landcover can be responsible for estimates of no emissions when trees are present or high emissions where trees are not present. These cases significantly affect the overall standard regression but the robust regression which uses bisquare weights gives a smaller weight to outliers and a higher weight to the points which are closer to the regression model. The MEGAN v.2.1 Landcover v.2.2 BEFs showed reasonable agreement for most ecoregions ($r_{standard\ fit}$=0.62, $r_{bisquare\ fit}$=0.89, slope 1.08 and no offset). The remaining ecoregions occur more or less equally in the region of model overestimation or underestimation. Overall the model BEF agrees with observed BEF within 10% which is substantially better than the stated 50% model uncertainty and the 20% measurement uncertainty that we estimated. The BEIGIS model BEFs are shown for comparison and they had good agreement for a smaller number of ecoregions and in many cases either significantly overestimated or underestimated the BEFs. However, overall the fit suggested about 30% of overestimation in BEFs and a small negative offset.

Interestingly, MEGAN v. 2.04 Landcover v.2.1 BEFs were characterized by similar total averages as MEGAN v.2.1 Landcover v.2.2 BEFs, but because of the smooth distribution of the BEF had fewer ecoregions matching measured BEFs as exactly as the other two landcovers although the discrepancies were also smoother with no extremes. The slope is only 0.56 but this is compensated by a very large positive offset of 1.35 mg m$^{-2}$ h$^{-1}$. As a consequence, the small BEF regions show overestimation of BEFs (e.g. in the Central Valley) but the high BEF regions tend to overestimate BEFs. In this case, the robust goodness of fit was not dramatically improved as was the case in the other two landcovers which had a much larger subset of ecoregions with explained variance. This comparison shows that each landcover could work relatively well for a global model, but clearly the latest landcover is most suitable for regional modeling. In any case, poorer agreement is expected for ecoregions where flight coverage was low or with extreme heterogeneity.



### 3.3 Comparison of CARB's hybrid model with CABERNET emissions

The primary goal of the study was to verify the accuracy of isoprene emission estimates used by CARB. For this reason, the emissions were simulated by CARB's hybrid model for exactly the same times and areas matching the CABERNET flux footprints to be compared with analogous 2-km measured emissions. Out of numerous simulations which were conducted between 4 km x 4 km and 1 km x 1 km resolutions and different footprint approaches, the best model-observation agreement was achieved for the 2 km x 2 km resolution and the most accurate footprints based on wavelet heat flux, wind speed and the ratio of altitude above the ground to planetary boundary layer depth ($z/zi$). In this paper we use non-directional symmetrical footprints. Upwind half-dome oriented footprints could be a better spatial approximation but are less practical in terms of the application to the existing CARB's modeling infrastructure. We determined that the full-dome approach we use for the homogenous oak woodlands should be similarly accurate except for a few areas at the boundaries of the oak woodland fetch or if there is a drastic inhomogeneity in landcover as indicated later in the analysis.

In Figure 7, the time series of simulated and measured emissions are shown to be generally in extremely good agreement (plotted along the complete flight tracks). Local discrepancies are observed in specific areas along the flight track and are discussed further in the next sections.

### 3.3.1 Sensitivity results

Modeled emissions are subject to uncertainties in the driving variables (temperature, PAR, LAI), so we performed sensitivity analyses to estimate their effect on the simulations.

#### *Temperature*

A $\pm$20% sensitivity analysis was done for the temperature input and showed that the measured emissions were within the range of modeled emissions for most of the dataset. The temperature dependence of isoprene emissions is exponential so the highest sensitivity is expected for higher temperatures. For example, at 20 °C 20% would correspond to a 4 °C difference while at 30 °C to a 6 °C difference. Because of the exponential character a 20% change in temperature could lead to changes in emissions as large as 100% above 30 °C. The highest errors in temperature used for simulations would be likely to occur in the areas close to the mountains where large gradients of temperatures (on the order of 10 °C) occur on the order of a few km and shift spatially during a day.

#### *PAR*

Similarly, a $\pm$20 % sensitivity analysis for the PAR input was tested in the model simulations. The resulting range of emissions was narrower than in the case of temperature sensitivity but the general picture was similar. A systematic offset in PAR (or temperature) would not improve significantly the generally excellent agreement, but it could improve or worsen the local agreement. For the cloudless skies during CABERNET it is unlikely that inhomogeneities in the spatial distribution of PAR could be significant although there could be an impact from an aerosol haze layer or high clouds in some areas.

#### *LAI*





The LAI and the cover fraction of oak woodlands can vary greatly in the Sierra foothills and it is expected that the LAI products from MODIS may not work ideally for oak landscapes. The MODIS LAI product is an average of all vegetation at a location and so would not discriminate for example between oak trees and grasses that occur together in oak woodlands. A ±50% uncertainty in LAI is therefore not unrealistic, thus we apply this uncertainty to the model and compare with the

5 measurements. This range in LAI resulted in relatively small changes in modeled emissions although occasionally substantial sensitivity to LAI was observed (even up to a factor of 2) but with no constant systematic offset. It is therefore assumed that the LAI used in the simulation was sufficiently accurate. The occasional model overestimations or underestimations were likely less related to the temperature (or LAI or PAR) than to the landcover inhomogeneity and inaccuracy.

### 3.3.2 Regional model performance over ecoregions

To test the regional performance of the model the data have been grouped over ecoregions and the resulting variabilities are shown independently for each of these ecoregions in the Supplement Figure 8. The direct comparison of measured vs modeled fluxes suggests agreement is remarkably good in most cases not only for the midrange from the statistical distribution but also in the case of episodic spatial events (e.g. see 6ai, 6b, 6r, and 6z). The direct flux comparison agrees generally quite well as with the BEF comparison approach earlier presented, but a few exceptions are apparent such as for 6ao and 6h. These two

subecoregions showed the highest discrepancy between the model and measurement but these two ecoregions were covered in less than 40 km of flight track, so are likely not statistically representative. The footprint integration can be an issue if the number of points for a given ecoregion is low so the inhomogeneity of the footprint could be the cause of the discrepancy. The high similarities between BEFs and fluxes in the remaining vast majority of subecoregions suggests that the footprint approach works well and shows that the CARB biogenic emission estimates agree generally well with observations and in many cases

including well covered and highly homogenous oak woodlands the agreement is excellent including the overall statistics (Table 1).

Although isoprene emissions were typically very low in the Central Valley, subecoregions 7m and 7o had considerable measured emissions which were not predicted by the model. These ecoregions correspond to the San Joaquin basin and Westside Alluvial Fans and Terraces, respectively, and the landcover database is likely missing isoprene sources which were within the

25 aircraft flux footprint but are not representative of the average for the entire subecoregion 7m or 7a. Another interesting observation is that the emissions simulated by CARB for flux footprint areas follow more closely the measured emissions, than the measured BEFs from the flights compared with BEFs averaged over entire ecoregions. Overall the BEF and area emission methods are consistent in their good agreement between measurement and model.

We quantitatively compare measured and modeled fluxes in Figure 8 (box plot statistics) and Figure 9 (scatter plot). Unlike

the BEF case which looked at BEFs averaged over entire ecoregions (of level IV) rather than for the corresponding areas of individual flux footprints, the $R^2$ is 0.96 with more than 70% of the points within the 95% confidence intervals. The 6h and 6ao ecoregion outliers are the most outstanding and have been discussed above. The lower emission graph shows that regions 5h, 6r, 6j, 6k, and 6z simulated emissions are overestimated. Region 5h is the Sierra Lower Mountain Forest ecoregion, and the other four are located in the northwestern coastal part of CA which is characterized by less homogenous coastal oak





terrains. This ecoregion could therefore be more sensitive to accuracies in spatial footprint positioning since some but not all of these overestimates were the case in the BEF comparison. This relatively small number of overestimates is balanced by underestimates (e.g. regions 7m, 7c, 14f, 6ag) where in some cases the modeled emissions were close to zero, suggesting inaccuracies of the landcover.

Approximately 30 ecoregions showing extremely good agreement demonstrate the emissions are accurately simulated based on the approaches we chose in these comparisons.

On average for the entire available flux dataset, we show that the model overestimates the emissions by 19% and this is driven by a few high episodic events in the simulations which were not observed in the measured emissions. Interestingly, when comparing the median values the model is also very close to the observation with 16% underestimation by the model.

This is excellent agreement which is much better than the predicted accuracy of either the modeled or measured values. The analysis points to the importance of regional assessments of the modeled emissions where in some cases discrepancies may occur.

For example, the subecoregion which was most extensively covered (~400 km, RF2, RF3, RF4) was 6b (Northern Sierran Foothills) and exhibited almost identical quantitative statistics for the model (mean 2.30, median 1.23, s.d. 2.66, min 0.008

and max 14.2 kg h-1), and measurements (mean 2.33, median 1.31, s.d. 2.67, min 0.000, and max 15.9 kg h$^{-1}$), and the qualitative correspondence suggests we should have high confidence in the combination of the wavelet flux measurement, footprint analysis, and the emission modeling approach. This ecoregion includes the most homogeneously distributed oak woodlands and is therefore perhaps easier to model correctly in terms of properly estimating isoprene emissions in CA.

Subecoregion 6d (Camanche Terraces) covered in 50 km of tracks was neighboring to the east with 6b and to the west with

7a, and with much sparser oaks showed lower emissions but still had reasonable agreement between the model (mean 0.364, median 0.113, s.d. 0.530, min 0.000, and max 1.70 kg h$^{-1}$) and measurements (mean 0.453, median 0.275, s.d. 0.440, min 0.000, and max 1.45 kg h$^{-1}$).

On the other hand, there are regions where quantitative agreement is less good, such as coastal 6ai (Interior Santa Lucia Range) represented in  400 km of the flight tracks where on average the model underestimated the emissions by approximately

a factor of two. Another example is subecoregion 7m (San Joaquin Basin), where the model showed zero emissions (over  50 km of tracks) and isoprene emissions were measured as high as 7.58 (mean 1.73) kg h$^{-1}$. An opposite example in a different region (6r, East Bay Hills/Western Diablo Range) had model overestimation by about a factor of 2. This region suffered from fires with the most notable fire storm in 1991. Apart from the changes in landcover, the discrepancies may be caused by inaccuracies in meteorological driving inputs although probably to a lesser degree based on results from our sensitivity study.

In a few cases at the boundary of the oaks the agreement may have been more sensitive to the full-dome flux footprint, but in majority of cases this footprint approach was sufficient to represent correctly the area sources. For highly heterogeneous areas a directional half-dome approach would work even better at finer scales.




## 4   Conclusions

Accurate prediction of isoprene emissions is crucial for atmospheric chemistry and air quality modeling in the state of California, as well as other forested regions around the world. We used direct airborne flux measurements over the main regions in California where emissions are expected to be high to evaluate CARB's emission estimates based on their new hybrid model

that is used for simulating isoprene emissions of those areas and is important for development of the state implementation plan (SIP) for air quality. The approaches that were used in the comparison of the model with observation involved comparison of airborne and landcover BEFs and independently the emissions integrated over the same footprint areas.

The overall agreement that was obtained was remarkably good. Mean measured and modeled emissions agreed within 50% for half of the ecoregions, while for 21% of the ecoregions the model overestimated mean measured emissions and for

29% the model underestimated emissions. On average the agreement of model with measurement was within 19% over the whole dataset. The conducted sensitivity tests for a 20% change in temperature, 20% change in PAR and 50% change in LAI altered the total mean of the simulated fluxes by up to 43%, 21%, and 40%, respectively, suggesting that these inputs are also important. Although the change in these input variables would not improve the overall agreement significantly, it could dramatically impact specific regional agreements.

The quality of the model output is directly tied to the input datasets and based on our analysis we conclude that the most important contributor to overall uncertainties in the input database is the landcover. While this was the first airborne regional evaluation of biogenic inventories for isoprene, the conclusion about the model landcover being the most important driving input is consistent with studies from other ecosystems which evaluated model landcovers (e.g. observations from Italian ecosystems (Pacheco et al., 2014) and other European ecosystems (Oderbolz et al., 2013). Future efforts should focus on de-

veloping highly resolved and highly accurate landcovers using a combination of airborne flux measurements, remote sensing data and other recently available tools.

*Acknowledgements.*   We gratefully acknowledge California Air Resources Board (CARB) for funding CABERNET Contract #09-339, and the CIRPAS team for help in instrument integration. We acknowledge Robin Weber and Abhinav Guha (UC Berkeley) for their contributions to the successful campaign. We would like to thank Steve Shertz (NCAR) for engineering support and Xiaoyan Jiang (NCAR) for assistance

with MEGAN and WRF simulations. NCAR is sponsored by the National Science Foundation. We also acknowledge Prof. Maggi Kelly at GIF, UC Berkeley for suggestions regarding geospatial landcovers.



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



**Table 1.** Summary quantitative statistics for CABERNET and CARB model's emissions (kg h$^{-1}$)*

| Ecoregion | Description | N | CABERNET | | | CARB MODEL | | |
|---|---|---|---|---|---|---|---|---|
| | | | Mean | Median | SD | Mean | Median | SD |
| Total | All ecoregions | 1746 | 1.38 | 0.416 | 2.74 | 1.64 | 0.360 | 4.34 |
| | | | | | Good agreement | | | |
| 5e | Northern Sierra Lower Montane Forests | 29 | 1.21 | 0.992 | 1.22 | 0.852 | 0.622 | 0.842 |
| 5h | Central Sierra Lower Montane Forests | 26 | 1.48 | 1.11 | 1.509 | 2.27 | 1.96 | 1.70 |
| 6aa | Eastern Hills | 28 | 0.113 | 0.000 | 0.231 | 0.095 | 0.026 | 0.216 |
| 6al | Salinas-Cholame Hills | 44 | 0.562 | 0.381 | 0.730 | 0.460 | 0.215 | 0.848 |
| 6ap | Solomon-Purisima-Santa Ynez Hills | 31 | 1.16 | 0.749 | 1.15 | 1.08 | 0.720 | 1.18 |
| 6b | Northern Sierran Foothills | 196 | 2.33 | 1.31 | 2.67 | 2.30 | 1.23 | 2.66 |
| 6c | Southern Sierran Foothills | 181 | 1.24 | 0.647 | 1.65 | 0.851 | 0.383 | 1.13 |
| 6d | Camanche Terraces | 24 | 0.453 | 0.275 | 0.440 | 0.364 | 0.113 | 0.530 |
| 6l | Napa-Sonoma-Russian River Valleys | 22 | 0.505 | 0.346 | 0.569 | 0.770 | 0.326 | 1.26 |
| 6z | Diablo Range | 136 | 0.944 | 0.252 | 1.88 | 1.70 | 0.592 | 2.66 |
| 7a | Northern Terraces | 27 | 0.266 | 0.130 | 0.365 | 0.182 | 0.074 | 0.262 |
| | | | | | Model underestimates | | | |
| 6ac | Temblor Range/Elk Hills | 36 | 0.073 | 0.037 | 0.093 | 0.000 | 0.00 | 0.00 |
| 6af | Salinas Valley | 24 | 0.223 | 0.00 | 0.341 | 0.140 | 0.040 | 0.214 |
| 6ag | Northern Santa Lucia Range | 30 | 4.09 | 1.05 | 5.47 | 1.22 | 0.607 | 1.39 |
| 6ai | Interior Santa Lucia Range | 201 | 2.83 | 1.17 | 4.41 | 1.24 | 0.307 | 2.92 |
| 6ak | Paso Robles Hills and Valleys | 36 | 0.927 | 0.513 | 1.24 | 0.453 | 0.108 | 0.975 |
| 6g | North Coast Range Eastern Slopes | 20 | 1.10 | 0.297 | 1.68 | 0.582 | 0.247 | 0.918 |
| 7j | Delta | 35 | 0.358 | 0.295 | 0.337 | 0.015 | 0.000 | 0.050 |
| 7m | San Joaquin Basin | 23 | 1.73 | 0.234 | 2.65 | 0.000 | 0.000 | 0.000 |
| 7o | Westside Alluvial Fans and Terraces | 38 | 0.683 | 0.203 | 0.994 | 0.004 | 0.000 | 0.014 |
| 7p | Gigantic Alluvial Fans and Terraces | 22 | 0.053 | 0.026 | 0.129 | 0.000 | 0.000 | 0.000 |
| 7t | South Valley Alluvium | 23 | 0.025 | 0.005 | 0.066 | 0.000 | 0.000 | 0.000 |
| | | | | | Model overestimates | | | |
| 6aj | Southern Santa Lucia Range | 23 | 0.665 | 0.205 | 0.820 | 4.72 | 2.59 | 4.84 |
| 6j | Mayacmas Mountains | 41 | 0.272 | 0.148 | 0.382 | 2.11 | 0.884 | 5.46 |
| 6k | Napa-Sonoma-Lake Volcanic Highlands | 22 | 1.241 | 0.423 | 1.80 | 6.86 | 1.92 | 12.7 |
| 6r | East Bay Hills/ Western Diablo Range | 204 | 1.516 | 0.388 | 3.06 | 3.87 | 0.854 | 6.80 |
| 78q | Outer North Coast Ranges | 32 | 1.040 | 0.297 | 1.64 | 4.67 | 1.32 | 10.8 |

*Ecoregions with N<20 (<40 km) were omitted from this table





**Figure 1.** USEPA Ecoregion map with overlaid CABERNET flight tracks covering most of code 6 ecoregions. The shapefiles used to produce the map in ArcGIS were downloaded from ftp://ftp.epa.gov/wed/ecoregions/ca/.

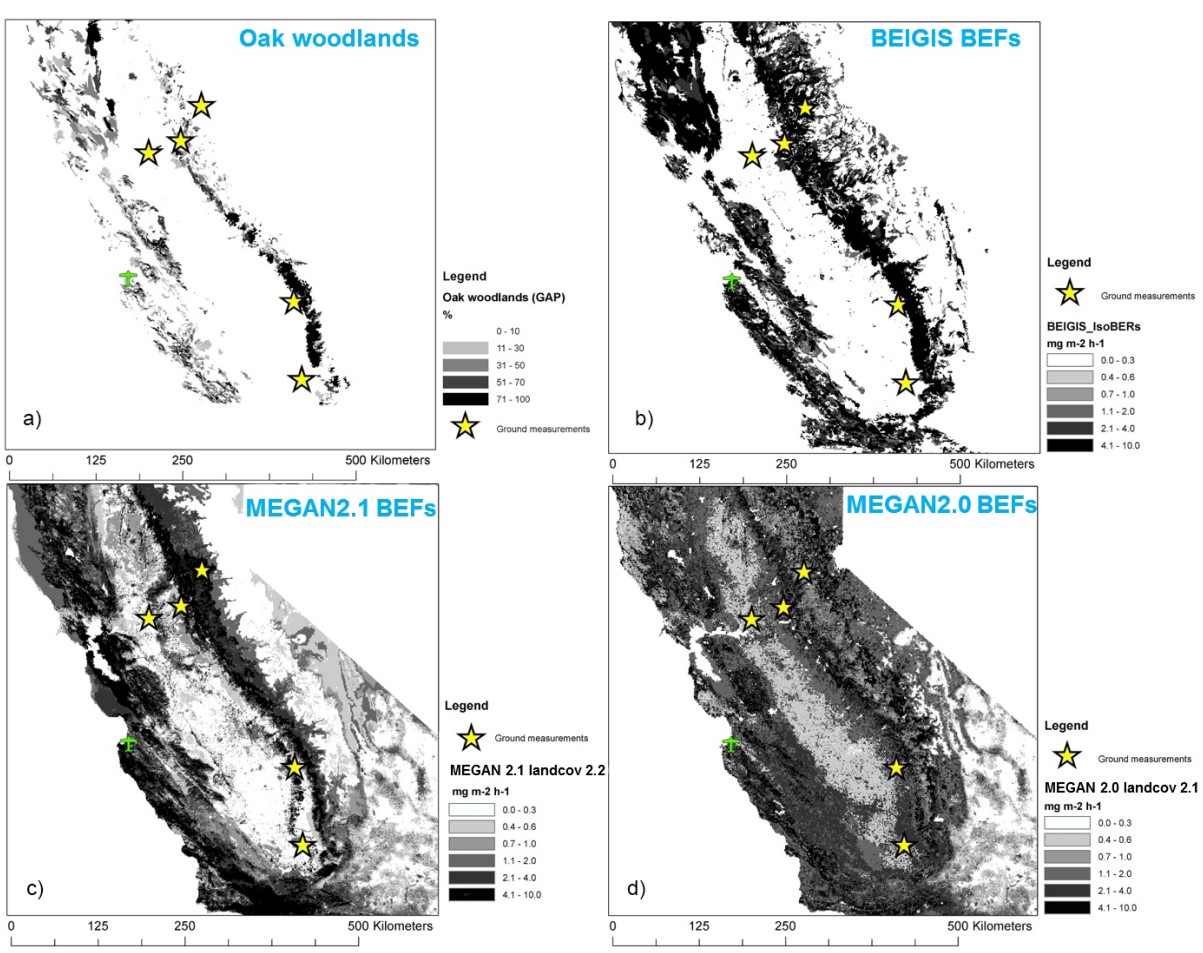

**Figure 2.** Landcovers used by the models. a) GAP's oak woodlands, b) BEIGIS emission factors (as dtiso+eiso) derived from the GAP database, c) MEGAN v.2.04 isoprene emission factors derived from landcover v.2.1, and d) MEGAN v.2.1 isoprene emission factors obtained from the most recent landcover v.2.2.





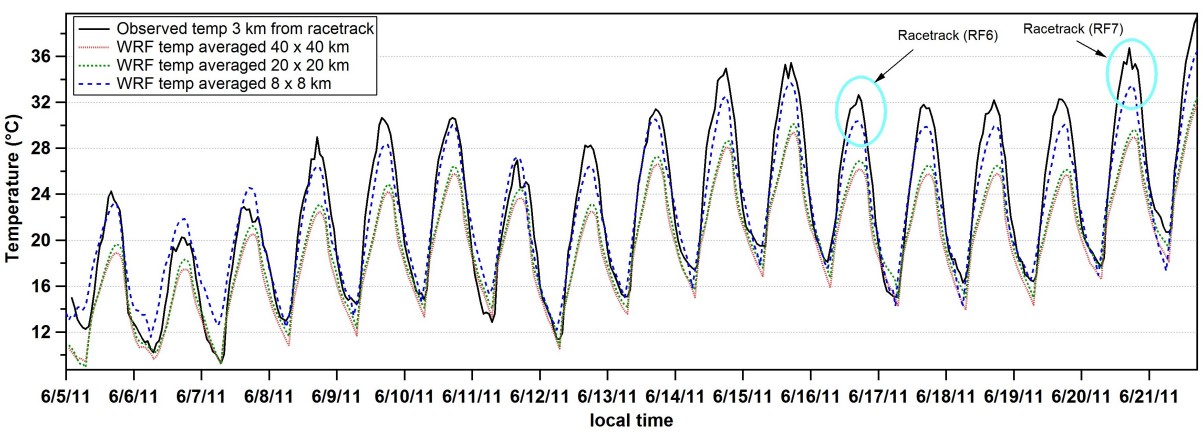

**Figure 3.** Resolution effect in WRF on temperature bias.



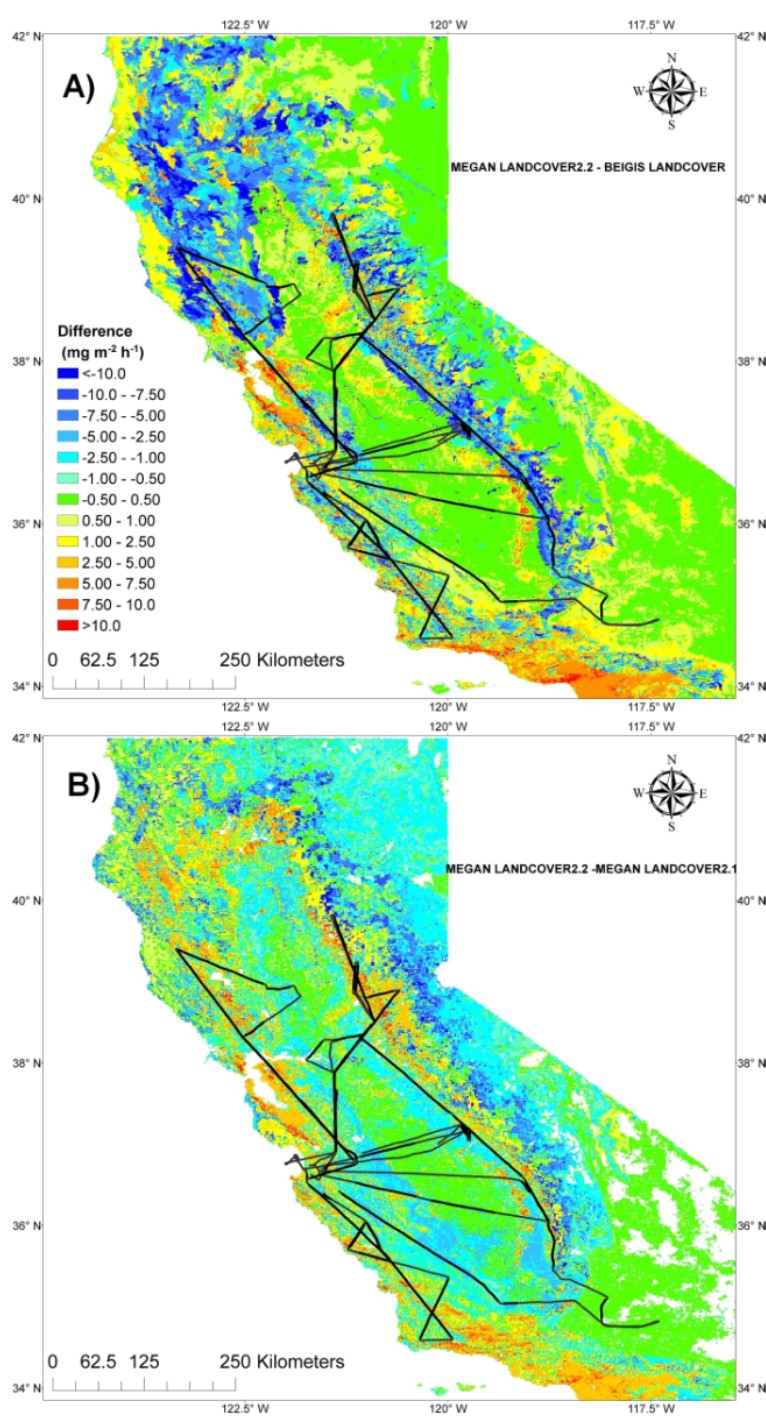

**Figure 4.** Absolute BEF differences of a) MEGAN v.2.1 Landcover v.2.2 and BEIGIS GAP Landcover and b) MEGAN v.2.1 Landcover v.2.2 and MEGAN v.2.04 Landcover v.2.1.





**Figure 5.** a) Comparison of airborne BEFs with MEGAN's landcover 2.2 for isoprene (airborne BEFs are subject to additional uncertainties introduced from $T$, and PAR used in normalization). Magnified areas are shown for b) northwest (including Northern Coastal Ranges to the left and Northern Sierra Foothills to the right, the middle area relates to the Central Valley and the San Joaquin Delta), c) central, and d) southeast tracks.





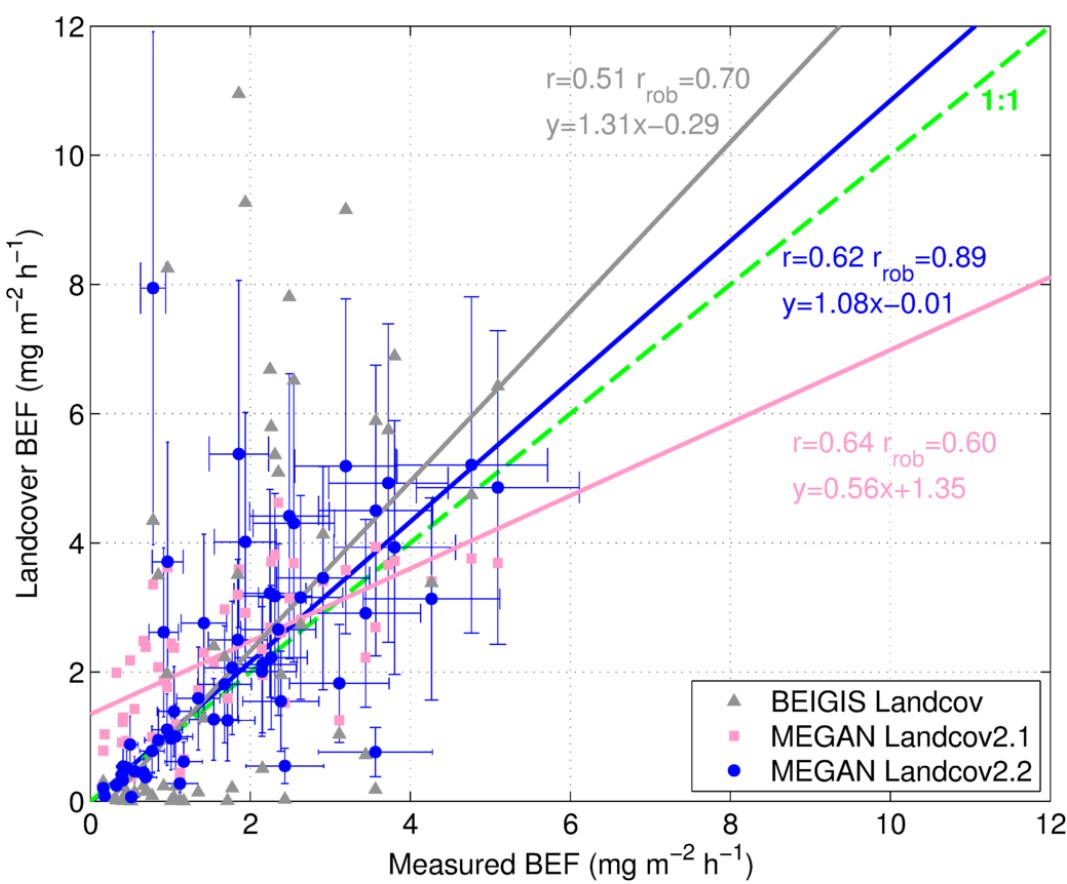

**Figure 6.** Comparison of measured versus modeled (MEGAN Landcover2.2, MEGAN Landcover 2.1, and BEIGIS) Basal Emission Factors averaged by USEPA ecoregion. Note: the number of averaged points in each ecoregion may be different and not necessarily representative of the entire ecoregion.







**Figure 7.** Time series for modeled and measured isoprene fluxes using the approximated circular footprint areas (only the data when flux was available are shown) along the full length of the flight tracks during the CABERNET campaign.





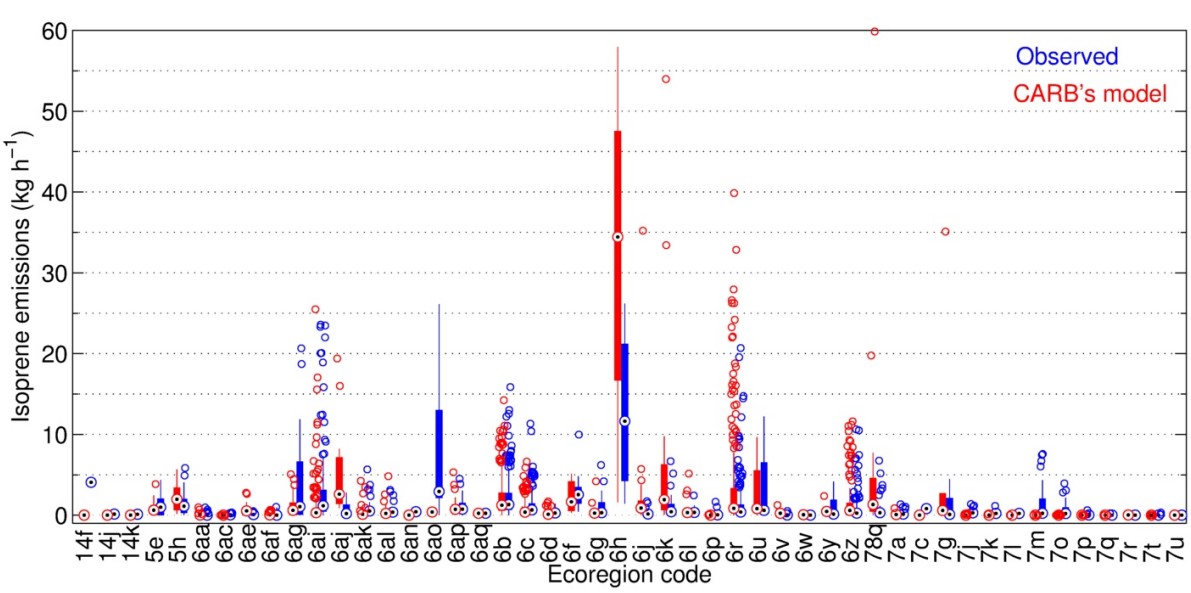

**Figure 8.** Box plots showing distribution of emissions in each of the level IV ecoregions. The boxes correspond to midrange (25th to 75th percentiles), the whiskers indicate variability outside the lower and upper quartiles, and the circles denote outlying emission hotspots.





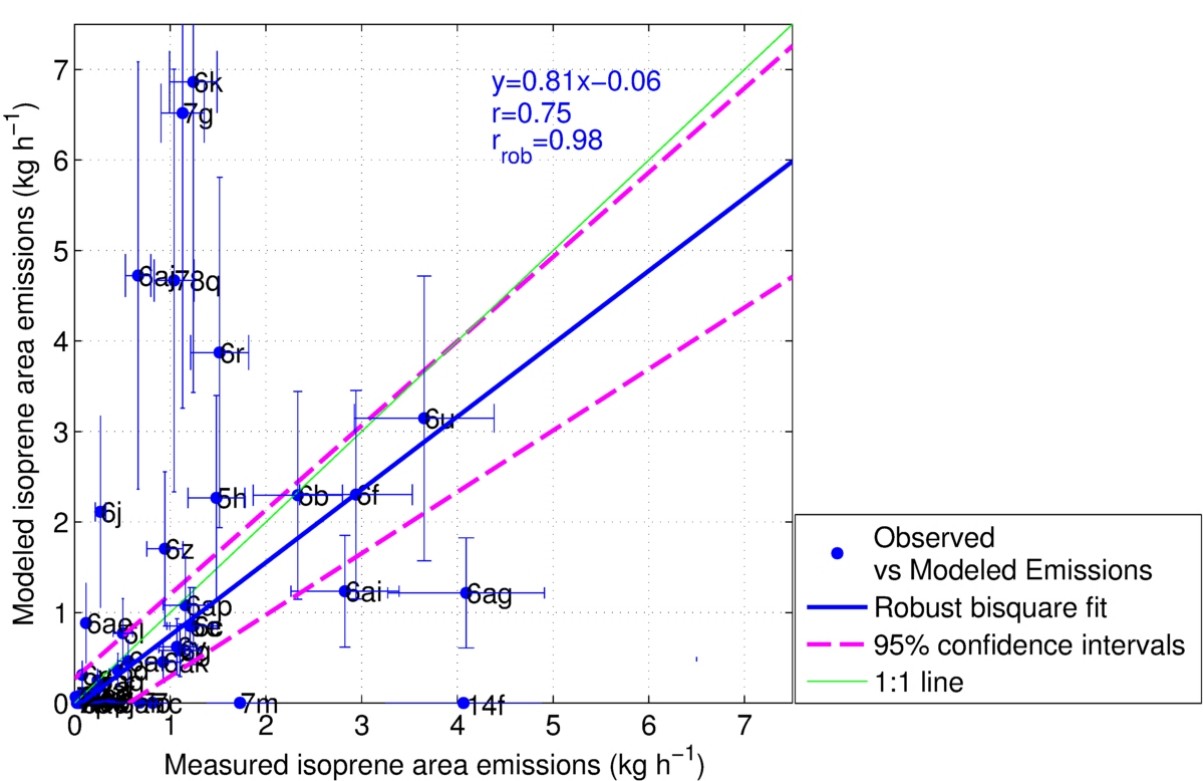

**Figure 9.** Scatter plot for the ecoregion averaged area emissions. The model dataset used is the hybrid CARB model. The vertical error bars represent the 50% model uncertainty and the horizontal error bars represent the 20% uncertainty of the measurement (applicable to ecoregions covered in more than 40 km – see Table 1).