# Peer review of "Evaluation of regional isoprene emission factors and modeled fluxes in California"

_Atmospheric Chemistry and Physics, 2016_

## Referee Comment (RC1) · Anonymous Referee #2 · 15 Apr 2016

Overview:

Based on airborne data from flight tracks over California, this study presents the evaluation of Basal Emission Factors, derived from measurements using different land-cover maps, and emission estimates, for the description of biogenic isoprene source. Land-covers considered in three different biogenic volatile organic compound emission models (BEIGIS, MEGAN 2.04 and MEGAN v.2.1) are considered for the calculation of Basal Emission Factors, and isoprene emission fluxes calculated by the CARB's hybrid model are evaluated.

This manuscript is well written and focuses on a very valuable and important work which totally falls in the scope of ACP topics. Biogenic source of volatile organic compounds are indeed still only crudely quantified, and model estimates associated with a

high uncertainty. Only a few studies presenting model-data comparison at a regional scale have been published so far, and I therefore both enjoyed the originality and the scientific contribution of this study and appreciated the work performed. However I strongly believe there is still a room for improvement in the presentation of this work in order to clarify the methodological approach and to present a deeper analysis of some of the results which are only quickly described. Here are some feedbacks on your manuscript and suggestions for improvements that I would really like to be considered before publication in ACP.

General comments:

The positioning and central objective(s) of this work have to be clarified, and homogenized in the manuscript. It is stated at the beginning of section 3.3 that "The primary goal of the study was to verify the accuracy of isoprene emission estimates used by CARB (. . .) simulated by CARB's hybrid model". And yet, very little space is eventually given in the manuscript to the evaluation of CARB's hybrid model results, especially if we do not consider the sensitivity tests for temperature, radiation and LAI, which to me are not really part of a model evaluation. This evaluation is presented as an independent work in the abstract, with only a few lines dedicated, while much more room is given to the BEF evaluation. All these different aspects of the work are really valuable and interesting to me, and are all worth being presented, but with the room entitled considering the main objectives given.

The agreement between measurements and model, regarding BEF and emissions, is somewhat overstated when described as "remarkable" (section 3.2.1) and "extremely good" (section 3.3). I agree that the main characteristic depicted by the measurements are generally well captured by the model (or model parameters i.e. BEF), which is already very encouraging considering the uncertainty in biogenic VOC emission estimates generally, but several emission peaks or BEF regional variability are still not captured by the model / model parameter. The comparison of isoprene fluxes simulated by CARB's hybrid model with measurements would really benefit from a deeper analysis:

what are the possible explanations for model/data disagreement regarding peak simulation, for instance? The objectives of the sensitivity tests should also be clarified: is the range of variability used for temperature/radiation representing the range of variability observed in the field? Moreover, regarding BEF especially, plots, rather than regional maps, comparing the BEF values for the same location along the different flight tracks would make the comparison analysis more visible.

It is stated, in the introduction and conclusion especially, that biogenic VOCs play a key role in California regarding air quality. What is exactly the contribution of biogenic sources to VOC emissions in this region? Please add some quantification.

Introduction, page 3, line 4: please clarify and detail "as well as the preceding meteorological history". Do you mean the past 24h or 10-days conditions for temperature and radiation, as taken into account in the MEGAN model?

Supplementary material and information are mentioned several times in the manuscript but unfortunately did not seem to be actually integrated in the supplementary document (which only presents 2 supplementary figures). So... either I missed a document well-hidden or some updates and corrections need to be performed. To properly understand the approach adopted for this work, it is indeed really important to find the information regarding "More methodological details" (page 8, line 15), "Further details on the application of the inverse algorithm" (page 8, line 28) and "Input variables tested" (page 9, line 21), supposed to be in a supplementary material, while Figure 8 does not seem to be part of this additional material anymore. Moreover the Figure S2, addressing the impact of fires on isoprene emissions, is indeed interesting but is not cited or used anywhere in the manuscript. I understand that supplementary information are not meant to be described in full details but at least should they be cited or used even shortly somewhere in the core of the manuscript, or deleted.

Many details, sometimes bringing a bit of confusion and that could be synthesized, are given regarding the different datasets used to build the landcover maps used by

the different models, but some of the differences and specificities of the models, which can affect significantly isoprene emission estimates, are not given clearly. For instance how many vegetation categories are considered by each model? How is LAI considered (grid-average or for each vegetation type) and which year or climatology is used? Adding a table summarizing all these information, together with model spatial and temporal resolutions, would really help. The CARB's hybrid model is described in section 2.1.4 as an adaptation of MEGAN v.2.04 to include MEGAN v.2.1 enhancements. So what are the actual differences between CARB's hybrid model and MEGAN v.2.1?

Section 3.1 and Figure 2: Landcover is shown to be a critical driving variable and yet only the oak woodland distribution is illustrated. It would be really worth and important to show the landcovers for all the different vegetation categories as well, simplified if needed depending on the number of vegetation categories, for each of the models' landcover, and not only the calculated BEF distribution.

Specific comments:

Page 1, line 5: change "Basal Emission Factors (BEFs) distribution" to "Basal Emission Factor (BEF) distribution".

Page 3, line 3: change "Large changes in temperatures" to "Large changes in temperature".

Page 4, line 5: add "to" in "this can contribute TO uncertainties in isoprene emission estimates".

Page 5, line 4: change "modeling with 1 km2" to "modeling with 1 km$^2$ ".

Page 5, line 28: change "MEGANv.2.1" to "MEGAN v.2.1", adding space.

The long name MEGAN v.2.1 landcover v.2.2 used several times in the manuscript could be shortened.

Tables and Figures:

Figure 1: The list of ecoregions is very long and hardly readable, and may not need to be presented in full details. This information should therefore either be simplified or enlarged/presented differently to be fully readable. If all these categories have to be presented, they could be listed in one table given in the supplementary material for instance.

Figure 2: Please check and correct the caption at this figure does not only give the "Landcovers used by the models", as stated at the beginning of the sentence, but also BEFs. Please also detail what "dtiso+eiso" means.

Figure 3 and section 2.3.2: The optimal approach regarding temperature accuracy was found to be using the 4 x 4 km WRF model nudged by CALMET or CALMET directly. This result is not illustrated in Figure 3, and should be added.

---

## Referee Comment (RC2) · Anonymous Referee #1 · 19 Apr 2016

The importance of accurately estimating isoprene emissions for simulating photochemistry and aerosol formation in chemical transport models is well recognized. Yet, after two decades, the uncertainty of isoprene emission estimates still hovers around a factor of two. If the results from this study are upheld, the bias will have been significantly improved, although there is still considerable variability between model and observations for specific events. This paper builds off efforts to reduce this uncertainty by analyzing "measured" airborne isoprene fluxes over a range of ecosystems (from low emitters to higher emitters) over a large portion of California. The usefulness of this paper hinges strongly on the airborne flux technique introduced by Misztal et al. (2014). Overall, the results of the paper should be of interest to the ACP reader community, and with modifications, the paper is recommended for publication in ACP.

It was unclear why three models (BEIGIS, MEGANv2.04, and MEGANv2.1) were eval-

uated with integrated ecoregion fluxes, while one model (CARB) was evaluated with footprint data. Because of the uncertainty in the size and shape of the footprints, it is recommended that the evaluation focus on all four models versus the integrated flux data across each ecosystem. Because of the importance of trying to establish reliable airborne derived footprints for a variety of other trace gases (such as $CO_2$, $CH_4$, and $N_2O$), maintaining the exploratory footprint section would be of interest to the community.

Specific comments:

Abstract: I am not a fan of abstracts written with paragraphs and references, but that is my personal preference. Suggest consulting ACP style guide to see if abstract follows protocol.

p.3, line 25: Since the CARB model was "improved" using the CABERNET measurements, evaluating it versus the other models using CABERNET measurements doesn't seem fair. Was any of the data set held out for an independent evaluation?

p.4, lines 6-11: Use of the WRF model to provide temperature and PAR is mentioned. And it is mentioned that the temperature data are compared against observations. How reliable is the PAR data? Because of aerosol effects, we have found that PAR from WRF can easily be overestimated by 10%.

p. 4, line 14: Sentence that begins with "BEIGIS" is fragmented and should be edited.

p. 5: Fundamentally, isoprene emissions from all four models depend on the normalized isoprene emission fluxes assumed for quercus. Either in this section or in the supplemental materials, please summarize the normalized isoprene fluxes assumed for each model, whether they are assumed at the leaf, branch, or canopy level, whether fluxes vary by quercus species, and any references to support the flux values. Similarly, isoprene fluxes depend on quercus leaf biomass densities. How do these vary among the four models? Perhaps, one way of comparing the three/four models would

be a table summarizing the attributes of the models that account for the differences in estimated isoprene fluxes. The narrative as currently written only provides a superficial insight into model differences.

p. 5, line 19 and other locations: In most situations, "which" should be preceded by a comma.

p. 5, section 2.1.4: Building on the "p. 5" comment above, more detail is needed to adequately describe the CARB hybrid model. Also, the sentence that begins "This regional model most closely agreed with the measured fluxes . . ." is confusing. My first reaction was why was a conclusion offered before the analysis and results sections. Then, I realized that the model had already been calibrated using the study data. This calibration needs to be clearly described here. All in all, this section is awkwardly written and needs to be reworked.

p. 8, section 2.4.1: The results of the paper rest on how the raw airborne data are converted into basal emission factors. The introductory paragraph mentions that more methodological details are provided in the supplement; however, these details are missing! These details are essential, but I didn't see them in the supplement. Lines 17-24, by themselves are insufficient, for convincing me of the accuracy of the airborne measurements in estimating basal fluxes. If possible, an estimate of errors associated with each of the six steps would be very helpful.

p. 9, lines 6-7: Clean up the font and subscripts for "dx0.5", "h", and "zm".

p. 13, line 7-8: Provide references and/or analysis to support the role of landcover in heterogeneity and inaccuracy in overestimates/underestimates. Perhaps uncertainties in the aircraft data and their translation could be a contributor?

p. 13, Section 3.3.2: In some of the tables and plots, emission amounts are used. Recommend sticking to fluxes. These could be area weighted to provide a perspective on the relative importance of different ecoregions.

p. 13, line 10: Add "," after "model".

p. 13, line 11: Remove "Supplement".

p. 13, line 20: After "woodlands", recommend showing specific ecoregion ids in parentheses.

p. 13, lines 22 and 25: Either ecoregion "7o" or "7a" is incorrect.

p. 13, line 23: Add comma after "emissions".

p. 13, lines 22-28: Given that high isoprene fluxes were measured over areas that seemingly have low isoprene emitting vegetation, have the authors considered the possibility of anthropogenic sources – such as petrochemical related facilities, like tire manufacturing?

p. 13, line 28: Add comma after "Overall".

p. 15, lines 2-7: While not necessarily a recommendation, this paragraph reads like it was extracted from a project report submitted to the state of California. Its style isn't consistent with most scientific journal articles.

p. 15, line 21: Be more specific on "other recently available tools".

Figure 1: Consider removing the portions of the flight tracks where flux data were unavailable.

Figure 2: "GAP's" should be defined. Plots c and d should be reversed to match caption.

Figure 3: The text should discuss the apparent underestimate of WRF for max temperature. There appears to be ∼2 deg C underestimate, which can strongly influence the Guenther estimates above 30 C.

Figure 4: Consider showing only those tracks with flux data.

Figure 7: This figure is confusing. It seems to imply a continuous range of footprint

sizes from 0 to 3500 km? Perhaps (if I understand correctly) it should be re-titled to read something like "distance along flight track". Also, why not use flux values rather than emission rates? I assume that 4 kg/hr = 1000 ug/m2-hr. For those portions of the flight track showing significant over- and under-estimates, why not be more specific with location in California and type of ecoregion(s)?

Figure 8: As mentioned earlier, recommend strongly that all four models be compared together. It is confusing to move from ecoregion averages to footprint emission amounts.

Supplement 1: As mentioned earlier, having more descriptions of the four models would be useful. Also, more details on how the basal emission fluxes were derived from the airborne measurements would be very helpful. See "p.8, section 2.4.1" comments above.

---

## Author Comment (AC1) · 7 Jun 2016

*We thank the reviewers for the very useful and insightful comments which improved the manuscript. The responses to each comment are shown in the italic text below, followed by the revised manuscript.*

Anonymous Referee #1

1) The importance of accurately estimating isoprene emissions for simulating photochemistry and aerosol formation in chemical transport models is well recognized. Yet, after two decades, the uncertainty of isoprene emission estimates still hovers around a factor of two. If the results from this study are upheld, the bias will have been significantly improved, although there is still considerable variability between model and observations for specific events. This paper builds off efforts to reduce this uncertainty by analyzing "measured" airborne isoprene fluxes over a range of ecosystems (from low emitters to higher emitters) over a large portion of California. The usefulness of this paper hinges strongly on the airborne flux technique introduced by Misztal et al. (2014). Overall, the results of the paper should be of interest to the ACP reader community, and with modifications, the paper is recommended for publication in ACP.

*We thank very much the reviewer for these very encouraging and favorable comments.*

2) It was unclear why three models (BEIGIS, MEGANv2.04, and MEGANv2.1) were evaluated with integrated ecoregion fluxes, while one model (CARB) was evaluated with footprint data. Because of the uncertainty in the size and shape of the footprints, it is recommended that the evaluation focus on all four models versus the integrated flux data across each ecosystem.

*We appreciate this suggestion but we chose to compare ecoregion averaged data instead consistently for all the models. This choice is motivated by the need for quantitative evaluation of the ecologically distinct regions in California which each may represent different modeling challenges. The advantage of our approach is uncertainty reduction from short term-variability as well as fine footprint uncertainties, which average out to a large extent when integrated spatially and temporally.*

*In addition to comparing by ecoregion average, we showed the performance and discussed the challenges of using discrete fluxes and footprints to inspire future research which could make progress in understanding smaller spatial scale variabilities. While for many tracks the agreement is very good even at this extremely fine resolution, the lack of agreement in some cases does not necessarily mean that the model does not work well, but rather that it is more difficult to account for the random errors at fine scale (e.g. 2 km) than with appropriate averaging (e.g. >40 km). In a future study it would be beneficial for the aircraft to fly numerous times over the same tracks to achieve higher confidence in resolving fine contributions to fluxes, but we did not have the luxury of doing such repeated flights during the CABERNET campaign, and chose instead to obtain larger spatial coverage.*

3) Because of the importance of trying to establish reliable airborne derived footprints for a variety of other trace gases (such as CO2, CH4, and N2O), maintaining the exploratory footprint section would be of interest to the community.

*In response to this comment, we expand the footprint application section (2.4.2) to include clarification of "full-dome", and "half-dome" footprints derived from wavelet analysis. While this has been the first application of these footprints, the development still continues. Another manuscript in preparation (Yu et al., in prep.) is focused on further refinement and application of wavelet footprint approaches and*

*recently Vaughan et al., 2016 compared emission inventory emission factors using aircraft wavelet NOx flux over London using a parameterized aircraft footprint model.*

Specific comments:

4)  Abstract: I am not a fan of abstracts written with paragraphs and references, but that is my personal preference. Suggest consulting ACP style guide to see if abstract follows protocol.

*We keep gentle paragraphing in the abstract (the spaces between paragraphs have been removed). The citation to Guenther algorithm is necessary to indicate the version. The citation to Misztal et al. (2014) makes it clear that the paper is not repeating the information which has been presented earlier.*

5)  p.3, line 25: Since the CARB model was "improved" using the CABERNET measurements, evaluating it versus the other models using CABERNET measurements doesn't seem fair. Was any of the data set held out for an independent evaluation?

*Both modeling and measurement efforts were occurring independently during and after CABERNET. The CARB model dataset has not been nudged yet in any way to measurement data so "was any data held out" does not apply. This way, the comparison is fair using the current CARB model which combines the latest developments in MEGAN but keeps improvements to regional BEIGIS infrastructure. Based on the comparison, there is still a scope to improve that model in different regions but the paper focuses on making fair comparison which will enable future improvements to the landcovers that the state might want to make. We make it now clear in the text that we did not use CABERNET fluxes to set the CARB emission factors.*

6)  p.4, lines 6-11: Use of the WRF model to provide temperature and PAR is mentioned. And it is mentioned that the temperature data are compared against observations. How reliable is the PAR data? Because of aerosol effects, we have found that PAR from WRF can easily be overestimated by 10%.

*This is a great point. We have only evaluated the temperature which can be more uncertain close to the foothills where gradients are larger. We also expect uncertainty in the PAR data but they are expected to be less prone to spatial differences relative to temperature and also will be small because we chose the flight days to be completely cloudless. However, we point out that averaging and aerosol loading can also have influence on PAR.*

7)  p. 4, line 14: Sentence that begins with "BEIGIS" is fragmented and should be edited.

*The sentence has been revised and now reads: "The Biogenic Emission Inventory processing model (BEIGIS) (Scott and Benjamin, 2003) was developed by CARB as a regional model specific to California, and is spatially resolved at 1 km² and temporally at 1 hour. BEIGIS uses California landcover, leaf mass, and emission rate databases with a geographic information system (GIS)."*

8)  p. 5: Fundamentally, isoprene emissions from all four models depend on the normalized isoprene emission fluxes assumed for quercus. Either in this section or in the supplemental materials, please summarize the normalized isoprene fluxes assumed for each model, whether they are assumed at the leaf, branch, or canopy level, whether fluxes vary by quercus species, and any references to support the flux values. Similarly, isoprene fluxes depend on quercus leaf biomass densities. How do these vary among the four models? Perhaps, one way of comparing the three/four models would be a table summarizing the attributes of the models that account for the

differences in estimated isoprene fluxes. The narrative as currently written only provides a superficial insight into model differences.

*We report ecosystem scale fluxes which are not normalized to mass but instead to land area, then corrected to environmental conditions to obtain basal emission factors on a per land area basis. The landcover basal emission factors used in the models include species independent emission factors which have been derived from leaf and branch level measurement scaled by leaf area. What we are comparing is the measured ecosystem scale flux derived Basal Emission Factors and the model landcover Basal Emission Factors.   This ecosystem scale comparison is unprecedented and consistently showed that these emissions were the highest from the oak woodlands which grow in specific altitude bands, but there were also substantial emissions in the mixed woodlands some of which could have been from other species such as Eucalyptus. As the reviewer suggests, we summarize the key attributes of the models, similarities and differences in the new table (Table 1).*

9) p. 5, line 19 and other locations: In most situations, "which" should be preceded by a comma.

*Done.*

10) p. 5, section 2.1.4: Building on the "p. 5" comment above, more detail is needed to adequately describe the CARB hybrid model. Also, the sentence that begins "This regional model most closely agreed with the measured fluxes . . ." is confusing. My first reaction was why was a conclusion offered before the analysis and results sections. Then, I realized that the model had already been calibrated using the study data. This calibration needs to be clearly described here. All in all, this section is awkwardly written and needs to be reworked.

*We apologize if the text gave the impression that the model had been calibrated. The model has not been calibrated on airborne data and was compared as it is. Initially we were considering to adapt the original MEGAN 2.1 which is currently the state-of-the-art version of the model but after a few pilot simulations it turned out that it would be difficult to keep previous regional enhancements in BEIGIS and it would be a major investment in modeling infrastructure change. After several preliminary pilot runs of the BEIGIS model with more enhancements from MEGAN 2.1, it was suggested that the hybrid model should perform similarly or even better than MEGAN 2.1 in CA regions. The sentence was to provide the explanation why the hybrid model was chosen and it has now been clarified to: "In preliminary runs (not shown), this regional model most closely agreed with the measured fluxes and is also currently used by CARB to estimate the BVOC emissions inventory for California. However, the model has not been calibrated on the measurement data to ensure that the comparison is fair."*

*As we mentioned in an earlier comment we summarize the detail for each model in the text and in Table 1. We have now improved the Sect. 2.1.4 in its clarity.*

11) p. 8, section 2.4.1: The results of the paper rest on how the raw airborne data are converted into basal emission factors. The introductory paragraph mentions that more methodological details are provided in the supplement; however, these details are missing! These details are essential, but I didn't see them in the supplement.

*We thank the reviewer for spotting this oversight in referencing the information which must have happened when we tried to move the specific text from the supplement to the manuscript. The methodological details on how airborne data are converted to basal emission factors were transferred from the supplementary information to the main manuscript and were available in Sect. 2.4.1 "Application of inverse G06 algorithm to the airborne fluxes". The text now correctly refers to Sect. 2.4.1*

*which has been further expanded to include the estimation of BEF uncertainty. In addition, the full equation of the algorithm with its parameters (default) is now shown in SI.*

12) Lines 17-24, by themselves are insufficient, for convincing me of the accuracy of the airborne measurements in estimating basal fluxes. If possible, an estimate of errors associated with each of the six steps would be very helpful.

*We add the estimates of errors in each step and discuss how they propagate to uncertainty in basal emission factors.*

13) p. 9, lines 6-7: Clean up the font and subscripts for "dx0.5", "h", and "zm".

*Done.*

14) p. 13, line 7-8: Provide references and/or analysis to support the role of landcover in heterogeneity and inaccuracy in overestimates/underestimates. Perhaps uncertainties in the aircraft data and their translation could be a contributor?

*In this particular sentence we did not mean to downplay any kind of uncertainties. They are all important to be aware of. The sentence was meant to say that spatial heterogeneity (not just in species composition, but also vegetation cover fraction) can be responsible for potential inaccuracy that is much larger than other uncertainties. The aircraft data have some uncertainty but inherently represents the heterogeneities which are much more difficult to represent accurately in any model. We included "likely" because it is difficult to determine precisely the magnitude of each element.*

15) p. 13, Section 3.3.2: In some of the tables and plots, emission amounts are used. Recommend sticking to fluxes. These could be area weighted to provide a perspective on the relative importance of different ecoregions.

*We assume that the reviewer suggests to stick with flux terminology, but as we do not expect deposition from isoprene, using the emission terminology makes sense and we do this consistently in the graphs and tables.*

16) p. 13, line 10: Add "," after "model".

*Done.*

17) p. 13, line 11: Remove "Supplement".

*Done.*

18) p. 13, line 20: After "woodlands", recommend showing specific ecoregion ids in parentheses.

*Done.*

19) p. 13, lines 22 and 25: Either ecoregion "7o" or "7a" is incorrect.

*7a was a typo. It has been changed to 7o.*

20) p. 13, line 23: Add comma after "emissions".

*Done.*

21) p. 13, lines 22-28: Given that high isoprene fluxes were measured over areas that seemingly have low isoprene emitting vegetation, have the authors considered the possibility of anthropogenic sources – such as petrochemical related facilities, like tire manufacturing?

*It is true that these areas in the Central Valley (7m, 7o) have little isoprene emitting vegetation in the model. However, the observed isoprene emissions were from vegetative regions, not from regions expected to have anthropogenic sources. We excluded significant anthropogenic isoprene contributions based on track analysis combined with Google Earth imagery. The aircraft observed gradual flux transitions characteristic of entering vegetative regions and the location of the vegetation was confirmed by Google Earth imagery.*

22) p. 13, line 28: Add comma after "Overall".

*Done.*

23) p. 15, lines 2-7: While not necessarily a recommendation, this paragraph reads like it was extracted from a project report submitted to the state of California. Its style isn't consistent with most scientific journal articles.

*We have revised these lines.*

24) p. 15, line 21: Be more specific on "other recently available tools".

*We added: "…such as highly sensitive time-of-flight mass spectrometry".*

25) Figure 1: Consider removing the portions of the flight tracks where flux data were unavailable.

*We appreciate the suggestion but we want to inform the reader where the track was, because even if the flux was not available the concentrations from those portions of tracks may still be valid.*

26) Figure 2: "GAP's" should be defined. Plots c and d should be reversed to match caption.

*GAP has been defined earlier in the text. We thank the reviewer for spotting the mismatch in the caption, which has now been corrected.*

27) Figure 3: The text should discuss the apparent underestimate of WRF for max temperature. There appears to be ~2 deg C underestimate, which can strongly influence the Guenther estimates above 30 C.

*We add a sentence to the caption and we discuss further these implications in Sect. 2.3.2.*

28) Figure 4: Consider showing only those tracks with flux data.

*As in response to comment 25, we want to show the full track where at least concentrations if not the flux were measured.*

29) Figure 7: This figure is confusing. It seems to imply a continuous range of footprint sizes from 0 to 3500 km? Perhaps (if I understand correctly) it should be re-titled to read something like "distance along flight track". Also, why not use flux values rather than emission rates? I assume that 4 kg/hr = 1000 ug/m2-hr. For those portions of the flight track showing significant over- and under-estimates, why not be more specific with location in California and type of ecoregion(s)?

*We agree that Figure 7 label might be confusing. The figure mostly served the purpose of showing overall comparison at finer scale along the tracks. The footprint is not constant but is derived for each point. We*

*show specific ecoregions on the next figures. We still thought it was worth leaving this figure in but in response to this and the next comment we decided to move this figure from the main manuscript to the supplement.*

    30) Figure 8: As mentioned earlier, recommend strongly that all four models be compared together. It is confusing to move from ecoregion averages to footprint emission amounts.

*In order to avoid confusion we deleted figure 7. All four models are compared as ecoregion averages.*

    31) Supplement 1: As mentioned earlier, having more descriptions of the four models would be useful. Also, more details on how the basal emission fluxes were derived from the airborne measurements would be very helpful. See "p.8, section 2.4.1" comments above.

*We really appreciate all the useful comments which we have implemented where possible. This was the first comparison of these models at regional scale and we hope future progress will continue in this area. We thank the reviewer for reading the manuscript carefully and for providing many useful comments.*

Anonymous Referee #2

    1) Based on airborne data from flight tracks over California, this study presents the evaluation of Basal Emission Factors, derived from measurements using different landcover maps, and emission estimates, for the description of biogenic isoprene source. Land-covers considered in three different biogenic volatile organic compound emission models (BEIGIS, MEGAN 2.04 and MEGAN v.2.1) are considered for the calculation of Basal Emission Factors, and isoprene emission fluxes calculated by the CARB's hybrid model are evaluated. This manuscript is well written and focuses on a very valuable and important work which totally falls in the scope of ACP topics. Biogenic source of volatile organic compounds are indeed still only crudely quantified, and model estimates associated with a high uncertainty. Only a few studies presenting model-data comparison at a regional scale have been published so far, and I therefore both enjoyed the originality and the scientific contribution of this study and appreciated the work performed.

*We really appreciate the positive feedback and in particular the compliment on "the originality and the scientific contribution of this study and appreciated the work performed".*

    2) However I strongly believe there is still a room for improvement in the presentation of this work in order to clarify the methodological approach and to present a deeper analysis of some of the results which are only quickly described. Here are some feedbacks on your manuscript and suggestions for improvements that I would really like to be considered before publication in ACP.

*Definitely, we realize there is much more to accomplish in this area and we thank the reviewer for highlighting the points worth attending to. We want to stress that the particular part of the study presented in the paper did not intend to focus on the methods (which are very interesting and were developed to accomplish the study) but rather on the scientific implications which could lead to improvements of model inputs at relevant temporal and spatial resolution for regional models.*

General comments:

    3) The positioning and central objective(s) of this work have to be clarified, and homogenized in the manuscript. It is stated at the beginning of section 3.3 that "The primary goal of the study was to

verify the accuracy of isoprene emission estimates used by CARB (. . .) simulated by CARB's hybrid model". And yet, very little space is eventually given in the manuscript to the evaluation of CARB's hybrid model results, especially if we do not consider the sensitivity tests for temperature, radiation and LAI, which to me are not really part of a model evaluation. This evaluation is presented as an independent work in the abstract, with only a few lines dedicated, while much more room is given to the BEF evaluation. All these different aspects of the work are really valuable and interesting to me, and are all worth being presented, but with the room entitled considering the main objectives given.

*The objectives are clarified in the revised version. The reviewer makes a great point that the sensitivities to input variables are not everything. We do think that the architectures of these models are quite similar and what we evaluate is the direct comparison to measured ecosystem variables. We put more emphasis on the CARB model in the revised version. We thank the reviewer very much for finding "all the different aspects of the work valuable, interesting and worth being presented".*

4) The agreement between measurements and model, regarding BEF and emissions, is somewhat overstated when described as "remarkable" (section 3.2.1) and "extremely good" (section 3.3). I agree that the main characteristic depicted by the measurements are generally well captured by the model (or model parameters i.e. BEF), which is already very encouraging considering the uncertainty in biogenic VOC emission estimates generally, but several emission peaks or BEF regional variability are still not captured by the model / model parameter. The comparison of isoprene fluxes simulated by CARB's hybrid model with measurements would really benefit from a deeper analysis: what are the possible explanations for model/data disagreement regarding peak simulation, for instance? The objectives of the sensitivity tests should also be clarified: is the range of variability used for temperature/radiation representing the range of variability observed in the field? Moreover, regarding BEF especially, plots, rather than regional maps, comparing the BEF values for the same location along the different flight tracks would make the comparison analysis more visible.

*We agree with the reviewer that we should allow the data speak more for themselves and not use descriptors such as extremely good or remarkable. We make a deeper insight into model agreements and disagreements. We also make it clear that the objective of the sensitivity studies was to represent the expected variability within realistic bounds.*

5) It is stated, in the introduction and conclusion especially, that biogenic VOCs play a key role in California regarding air quality. What is exactly the contribution of biogenic sources to VOC emissions in this region? Please add some quantification.

*We now add in the introduction: "In CARB's current emission inventory (CARB 2015), biogenic sources constitute 60% of total VOC emissions in California. Isoprene accounts for 37% of the biogenic VOC and 22% of total VOC. Furthermore, the important impacts of isoprene and other biogenic VOC emissions on total VOC reactivity, ozone formation, and aerosol formation in the Central Valley and surrounding mountains have been demonstrated in many previous studies (Kleinman et al., 2016; Worton et al., 2013; Rollins et al., 2012; Steiner et al., 2008; Dreyfus et al., 2002) pointing to the need for assessing the accuracy of emission inventories." (…) "In this work we focus on quantifying the agreement between observed and modeled isoprene emissions from its main sources as an important step leading to increased confidence in air quality predictions."*

6) Introduction, page 3, line 4: please clarify and detail "as well as the preceding meteorological history". Do you mean the past 24h or 10-days conditions for temperature and radiation, as taken into account in the MEGAN model?

*Yes, we meant to draw attention to the transition from a relatively cool period to a relatively warmer period of early summer, which was responsible for the wide range of conditions. Therefore, 24 and 240 h preceding meteorological history would be more relevant than during the middle of the season.*

7) Supplementary material and information are mentioned several times in the manuscript but unfortunately did not seem to be actually integrated in the supplementary document (which only presents 2 supplementary figures). So. . . either I missed a document wellhidden or some updates and corrections need to be performed. To properly understand the approach adopted for this work, it is indeed really important to find the information regarding "More methodological details" (page 8, line 15), "Further details on the application of the inverse algorithm" (page 8, line 28) and "Input variables tested" (page 9, line 21), supposed to be in a supplementary material, while Figure 8 does not seem to be part of this additional material anymore. Moreover the Figure S2, addressing the impact of fires on isoprene emissions, is indeed interesting but is not cited or used anywhere in the manuscript. I understand that supplementary information are not meant to be described in full details but at least should they be cited or used even shortly somewhere in the core of the manuscript, or deleted.

*We thank the reviewer for pointing out these issues all of which are fixed in the revised manuscript. As we mentioned in response to a similar comment from Reviewer 1, the referenced text in the supplement was not transferred properly and this has now been fixed. We delete the figure showing the previous fire history in some regions containing oak woodlands.*

8) Many details, sometimes bringing a bit of confusion and that could be synthesized, are given regarding the different datasets used to build the landcover maps used by the different models, but some of the differences and specificities of the models, which can affect significantly isoprene emission estimates, are not given clearly. For instance how many vegetation categories are considered by each model? How is LAI considered (grid-average or for each vegetation type) and which year or climatology is used? Adding a table summarizing all these information, together with model spatial and temporal resolutions, would really help. The CARB's hybrid model is described in section 2.1.4 as an adaptation of MEGAN v.2.04 to include MEGAN v.2.1 enhancements. So what are the actual differences between CARB's hybrid model and MEGAN v.2.1?

*These are all very good points. In response to this and Reviewer 1 comment #12 we add a new Table 2 with a summary describing the main attributes of each model. The models use explicit emission factors which do not contain information about vegetation categories, although it is true that each of these models can have any number of PFTs as well as a specific number(s) of the default PFTs.*

9) Section 3.1 and Figure 2: Landcover is shown to be a critical driving variable and yet only the oak woodland distribution is illustrated. It would be really worth and important to show the landcovers for all the different vegetation categories as well, simplified if needed depending on the number of vegetation categories, for each of the models' landcover, and not only the calculated BEF distribution.

*We appreciate this comment but we did not use any PFT categories that make up the emission factors but instead used the explicit emission factors for the specific model applications in the manuscript. We do*

*agree that for places other than California and for global models it is indeed important to show different vegetation categories. While we emphasize that oak woodlands are extremely high isoprene emitters dominating isoprene sources in California we do note other isoprene emitters are present such as Eucalyptus trees and other less significant sources.*

Specific comments:

10) Page 1, line 5: change "Basal Emission Factors (BEFs) distribution" to "Basal Emission Factor (BEF) distribution".

*Done.*

11) Page 3, line 3: change "Large changes in temperatures" to "Large changes in temperature".

*Done.*

12) Page 4, line 5: add "to" in "this can contribute TO uncertainties in isoprene emission estimates.

*Done.*

13) Page 5, line 4: change "modeling with 1 km2" to "modeling with 1 km$^2$".

*Done.*

14) Page 5, line 28: change "MEGANv.2.1" to "MEGAN v.2.1", adding space. The long name MEGAN v.2.1 landcover v.2.2 used several times in the manuscript could be shortened.

*The space has been added. The long name is needed to avoid potential confusion with the version of MEGAN and landcover".*

Tables and Figures:

15) Figure 1: The list of ecoregions is very long and hardly readable, and may not need to be presented in full details. This information should therefore either be simplified or enlarged/presented differently to be fully readable. If all these categories have to be presented, they could be listed in one table given in the supplementary material for instance.

*The figure legend containing specific names of the ecoregions has been enlarged and moved to the supplement.*

16) Figure 2: Please check and correct the caption at this figure does not only give the "Landcovers used by the models", as stated at the beginning of the sentence, but also BEFs. Please also detail what "dtiso+eiso" means.

*The caption has been updated accordingly to describe "BEF landcovers". dtiso and eiso are the components of BEIGIS EFs and represent isoprene emission factor for deciduous and evergreen trees, respectively. This information has also been added.*

17) Figure 3 and section 2.3.2: The optimal approach regarding temperature accuracy was found to be using the 4 x 4 km WRF model nudged by CALMET or CALMET directly. This result is not illustrated in Figure 3, and should be added.

*Unfortunately, we have not made 4 x 4 km runs for this domain which was done before the CARB's model development. However, 8 x 8 km already worked pretty well. The CALMET dataset has been, in addition, independently evaluated by CARB to ensure its accuracy. Consistently, however, the accuracy must necessarily be less good near the foothills which we have highlighted in the manuscript.*

*Once again, we thank the reviewer for providing thoughtful comments which we found very useful for the improvement of our manuscript.*

Yu, H., et al.: Development of landcover and emission factors for isoprene and monoterpene emission modeling and evaluation in the southern United States using airborne direct and indirect flux measurements, in prep.

[revised manuscript text omitted]

**S1. Ecoregion codes (Legend to Figure 1)**

[Figure]

*Figure S1. Legend to Figure 1 describing ecoregion codes.*

**S2. MEGAN architecture and main differences between versions**

The main differences of MEGAN v.2.1 to MEGAN v.2.04 are:

1) v2.04 does not have soil moisture or CO2 response (but these were not used for MEGAN v.2.1 simulations in this study);

2) MEGAN v.2.04 uses a different emission factor database and has different light response algorithms (which are nearly the same for isoprene and mostly impact other compounds);

3) MEGAN v.2.04 uses different parameters in the canopy environment model.

[Figure]

*Figure S2. Schematic of MEGAN v.2.1 model components and driving variables (taken from Guenther et al., 2012).*

**S3. Timeseries of simulated and observed emissions**

In Figure S3, the time series of simulated and measured emissions are shown (plotted along the complete flight tracks).

Local similarities and discrepancies are observed in specific areas along the flight track and are discussed in the manuscript. Although there are different sources of uncertainty, the largest discrepancy occurs if the trees are significantly under or overrepresented, which could be due to fires, new growth, or incomplete landcover.

[Figure]

*Figure S3. Time series for modeled and measured isoprene fluxes using the approximated circular footprint areas (only the data when flux was available are shown) along the full length of the flight tracks during the CABERNET campaign.*

**S4. The inverse G06 algorithm used in airborne emission factor derivation**

In the original G06 algorithm (equation below), $F_{G06}$ is the unknown, and BER is the known emission factor at standard temperature and PAR conditions. We inverse the equation so the BER is unknown and $F$ is the airborne-derived surface flux. This BER is referred to as airborne basal emission factor (BEF) or just emission factor which represents the airborne flux inferred for the standard conditions of PAR=1000 µmol m$^{-2}$ s$^{-1}$ and temperature = 30 °C.

$$F_{G06} = \underbrace{BER \cdot b_3 \cdot \exp\left[b_2 \cdot (P_{24} - P_0)\right] \cdot (P_{240})^{0.6} \cdot \frac{[b_1 - b_2 \ln(P_{240})] \cdot PAR}{\sqrt{1 + [b_1 - b_2 \ln(P_{240})]^2 \cdot PAR^2}}}_{\gamma_P} \cdot \underbrace{b_5 \cdot \exp\left[b_6 \cdot (T_{24} - 297)\right] \cdot \exp\left[b_6 \cdot (T_{240} - 297)\right] \cdot \frac{C_{T2} \cdot \exp\left[C_{T1} \cdot \left(\frac{1}{T_{opt}} - \frac{1}{T}\right) \cdot \frac{1}{0.00831}\right]}{C_{T2} - C_{T1} \cdot \left[1 - \exp\left(C_{T2} \cdot \left(\frac{1}{T_{opt}} - \frac{1}{T}\right) \cdot \frac{1}{0.00831}\right)\right]}}_{\gamma_T}$$

The micrometeorological variables include temperature close to the surface ($T$) and PAR. Previous 24 and 240-hour history of temperature and PAR are accounted for in $T_{24}$, $P_{24}$, $T_{240}$, $P_{240}$ variables. The parameters of the algorithm were used as default (i.e. $C_{T1}$=95, $C_{T2}$=230, $T_b$=313, $P_0$=200, $b_1$=0.004, $b_2$ = 0.0005, $b_3$=0.0468, $b_4$=0.6, $b_5$=2.034, $b_6$=0.05).

**Supplementary references:**

Guenther, A. B., Jiang, X., Heald, C. L., Sakulyanontvittaya, T., Duhl, T., Emmons, L. K., and Wang, X.: The Model of Emissions of Gases and Aerosols from Nature version 2.1 (MEGAN2.1): an extended and updated framework for modeling biogenic emissions, Geosci Model Dev, 5, 1471-1492, 10.5194/gmd-5-1471-2012, 2012.